# DEAL: Diffusion Evolution Adversarial Learning for Sim-to-Real Transfer

**Wentao Xu, Huiqiao Fu**\*, **Haoyu Dong, Zhehao Zhou, Chunlin Chen**\*
Department of Control Science and Intelligent Engineering,
School of Management and Engineering, Nanjing University, China.
{wentaoxu, haoyudong, zhzhou}@smail.nju.edu.cn,
hqfu@smail.nju.edu.cn, clchen@nju.edu.cn

## Abstract

Training Reinforcement Learning (RL) controllers in simulation offers cost-efficiency and safety advantages. However, the resultant policies often suffer significant performance degradation during real-world deployment due to the reality gap. Previous works like System Identification (Sys-Id) have attempted to bridge this discrepancy by improving simulator fidelity, but encounter challenges including the collapse of high-dimensional parameter identification, low identification accuracy, and unstable convergence dynamics. To address these challenges, we propose a novel Sys-Id framework that combines Diffusion Evolution with Adversarial Learning (DEAL) to iteratively infer physical parameters with limited real-world data, which makes the state transitions between simulation and reality as similar as possible. Specifically, our method iteratively refines physical parameters through a dual mechanism: a discriminator network evaluates the similarity of state transitions between parameterized simulations and target environment as fitness guidance, while diffusion evolution adaptively modulates noise prediction and denoising processes to optimize parameter distributions. We validate DEAL in both simulated and real-world environments. Compared to baseline methods, DEAL demonstrates state-of-the-art stability and identification accuracy in high-dimensional parameter identification tasks, and significantly enhances sim-to-real transfer performance while requiring minimal real-world data.

## 1 Introduction

In recent years, Reinforcement Learning (RL) controllers have achieved significant success in robotic control through iterative trial-and-error learning [1–5]. This enables robust control of complex robots even in unstructured environments. However, such learning processes demand extensive data collection, and conducting training directly in the real world poses challenges related to efficiency and safety. Simulation-based training offers a practical alternative by enabling large-scale data generation in parallelized and risk-free environments [6]. Despite this, policies trained in simulation often experience substantial performance degradation when deployed on real-world systems due to the sim-to-real gap.

To mitigate the performance degradation caused by the sim-to-real gap, previous research has explored various sim-to-real transfer techniques [7]. Among these, Domain Randomization (DR) is one of the most widely used approaches [8–12]. It improves policy robustness by randomizing environmental and robotic physical parameters within predefined ranges during training and injecting noise into observations and actions. This paradigm aims to expose agents to sufficiently diverse simulated scenarios, thereby enhancing robustness and facilitating zero-shot transfer to reality. However, DR

---

\*Corresponding authors

often relies heavily on expert knowledge to define suitable randomization ranges, tends to produce overly conservative behaviors [13], and may lead to unstable or prolonged training. Recent advances in simulator fidelity optimization [14–22] have introduced few-shot approaches that leverage limited real-world data to learn target environment parameter distributions, thereby narrowing the reality gap and aligning simulator physical dynamics with the reality. Nevertheless, these approaches still encounter challenges including high-dimensional system identification collapse, low identification accuracy, inherent instability during the alignment process.

To address these challenges, we introduce a novel Sys-Id framework for sim-to-real transfer, which innovatively combines Diffusion Evolution with Adversarial Learning (DEAL) to narrow the reality gap. As diffusion models have demonstrated exceptional capabilities in image and video synthesis [23–27], existing works primarily utilize them as high-dimensional data generators. In DEAL, Diffusion Evolution (DE) [28] is employed both as a high-dimensional optimizer for parameter search and as a powerful generator within a Generative Adversarial Network (GAN) [29]. This dual role enables the generation of physical parameters that bring simulated state transitions closer to those observed in the real world. Meanwhile, a discriminator network evaluates each evolving simulator by scoring its fitness based on the similarity between simulated state transitions and real-world demonstrations, thereby guiding the evolution of parameter distributions. DEAL builds upon this adversarial learning framework but avoids the need for costly real-world data to train a noise prediction module, as required in traditional diffusion models. Instead, it estimates optimal parameters by weighting the entire parameter population according to their fitness scores, predicts noise based on the current parameters and the estimated optimum, and approaches the target parameters through an iterative denoising process. To further improve performance, DEAL introduces automatic noise adaptation to balance exploration and exploitation and applies parameter normalization to reduce crossover errors caused by parameter scale imbalances. These enhancements enable DEAL to achieve stable and accurate high-dimensional parameter identification, significantly improving sim-to-real transfer with minimal real-world data and reduced computational cost.

In the experiments, we evaluate DEAL on five sim-to-sim tasks (AllegroHand, Humanoid, Go2, Cartpole, Ant) and two sim-to-real tasks (Cartpole, Go2). First, we evaluate DEAL's parameter identification capabilities, particularly in high-dimensional settings. Using a policy trained with Uniform Domain Randomization (UDR), we collect demonstrations in target environment and conduct parameter searches with DEAL to redefine the simulator. The policy is then retrained in the enhanced simulator and its transfer performance is tested in the target domain. We further assess DEAL's adaptability by expanding the search scale and analyzing its dependence on target-domain demonstrations. Finally, we complete the challenging sim-to-real transfer task. Experimental results show that DEAL achieves state-of-the-art stability and identification accuracy in high-dimensional parameter identification tasks, effectively bridging the sim-to-real gap with limited real-world data. In particular, the contributions of this work are threefold:

1. We introduce a novel methods, DEAL, which innovatively combines Diffusion Evolution with Adversarial Learning to narrow the reality gap.

2. We develop a automatic noise adaptation mechanism to balance exploration and exploitation, and propose a parameter normalization framework to counteract search errors caused by parameter scale imbalances.

3. We demonstrate the effectiveness of DEAL in both sim-to-sim and sim-to-real tasks, showing superior performance compared to existing baselines.

## 2   Related Work

Transferring simulation-trained policies to the reality in a stable and cost-effective manner has long been a goal for sim-to-real research. Previous work has approached sim-to-real transfer from both policy adaptation and system identification perspectives.

Policy adaptation optimization focuses on training policies that are robust to dynamics discrepancies without relying on real-world fine-tuning. Domain randomization (DR) is a foundational technique that injects randomness into simulation parameters to encourage generalization [8–10, 13]. However, DR often requires manual tuning of randomization ranges, leading to suboptimal performance or excessive training cost. Recent advancements mitigate these limitations by integrating curriculum learning [30] or GAN-guided subspace prioritization [31], Dropo further improves upon DR by

leveraging off-policy data or human demonstrations to learn a more effective randomization distribution [11]. Other methods introduce latent adaptation modules [32, 33] and employ Concurrent Policy Optimization [34] to estimate environment variables in real time, but these approaches struggle with high-dimensional or partially observable dynamics. Domain adaptation techniques, such as image-to-image translation [35], grounded action transformation (GAT) [36–38] and ASAP [39], aim to align simulation-reality action spaces or visual inputs. However, they typically require costly real-world data and often lack generalizability across tasks.

System identification optimization calibrates simulation parameters to match real-world dynamics using limited real-world trajectories. Inspired by classical Sys-ID frameworks [40–42], recent work has focused on identifying simulation parameters that better align with real-world dynamics. Bayesian approaches like BayesSim [21] and its extensions [22, 43] formalize system identification as an inference problem, iteratively estimating posterior distributions over simulation parameters. However, these methods suffer from high computational costs and poor scalability in high-dimensional parameter spaces. Some data-efficient approaches adopt alternative strategies, ASID [19] designs exploration policies to collect informative real-world trajectories, while TuneNet [15] employs supervised learning to map simulation trajectories to parameter gradients, bypassing iterative optimization. RL-based methods reframe the parameter search problem as a policy learning task, where trajectories serve as states and parameters as actions [17, 44], these methods struggle with sparse rewards in complex tasks. For dynamics alignment, adversarial training frameworks like SimGAN [18] and EASI [14] utilize a discriminator to distinguish simulated and real-world state transitions. EASI optimizes by selecting fixed elite parameters through manual sorting in evolutionary search, ignoring the information of the entire population. This reduces parameter population diversity and leads to a tendency to get trapped in local optima, which limits both the amount and accuracy of the searched parameters and often produces multiple sub-optimal solutions in system identification. In contrast, DEAL employs soft probabilistic weighting over the entire parameter population, preserving diversity and improving convergence stability. Its iterative denoising process continuously refines parameters without relying on a noise prediction module, enabling stable and precise high-dimensional parameter search with minimal real-world demonstrations and lower computation cost, while effectively enhancing transfer capabilities in complex robotic tasks.

## 3 Approach

In this section, we first provide a brief background to DE and GAN, then describe in detail how DEAL integrates DE with GAN, and demonstrate how it narrow the reality gap by aligning simulator's physcial parameters using limited real-world data.

### 3.1 Background

**Diffusion Evolution** Diffusion models [24, 25] operate in two phases: diffusion and denoising. In the diffusion phase, Gaussian noise is progressively added to the original data, while a noise prediction module $\epsilon_\theta$ is trained to predict the added noise. The denoising phase iteratively estimates the original data and performs directed denoising to recover it. Diffusion Evolution [28] (DE) reinterprets this framework through evolutionary principles: denoising mimics evolution, diffusion emulate reversed evolution and add random perturbations act as mutations. This algorithm uses the iterative denoising process from diffusion models to refine solutions in a parameter space, instead of recovering a data distribution, it shifts an initial random population toward an optimized solution distribution. Similar to the relationship between energy and probability in statistical physics, evolutionary tasks can be connected to generative tasks by mapping fitness to probability density, diffusion models are directly predicting the original data $x_0$ from noisy versions of $x_0$ at each time step, DE can estimate the optimal point by weighting the current population individuals $x$ based on the corresponding fitness probabilities:

$$\hat{x}_0(x_t, \alpha, t) = \frac{1}{Z} \sum_{x \in X_t} g[f(x)] \mathcal{N}(x_t; \sqrt{\alpha_t} x, 1 - \alpha_t) x, \tag{1}$$

where $x_t$ is the corrupted data at timestep $t \sim [0, T]$, diffusion schedule $\alpha_t$ governs the noise , $x$ is a sample from the current data distribution $X_t$, the probability mapping function $g(\cdot)$, typically implemented as Softmax, transforms fitness evaluated by $f(\cdot)$ into probabilities, and the Gaussian distribution indicates the conditional probability of the current data point given any sample point as

the optimum, the weighted results of these are normalized by $Z$. Given the design of the diffusion process, i.e., $x_t = \sqrt{\alpha_t}x_0 + \sqrt{1-\alpha_t}\epsilon$, the noise $\epsilon$ can be estimated without the need for noise prediction module $\epsilon_\theta$ by:

$$\hat{\epsilon}(x_t, \alpha, t) = \frac{x_t - \sqrt{\alpha_t}\,\hat{x}_0(x_t, \alpha, t)}{\sqrt{1-\alpha_t}}. \tag{2}$$

Under DDIM [25] framework, this step-by-step denoising process can be described as:

$$x_{t-1} = \sqrt{\alpha_{t-1}}\hat{x}_0 + \sqrt{1-\alpha_{t-1}-\sigma_t^2}\hat{\epsilon} + \sigma_t\omega, \tag{3}$$

where $\omega$ is the random perturbations controlled by the noise schedule $\sigma_t$ in denoising phase. More proofs about DE can be found in the Appendix A.1.

**Generative Adversarial Network**   The Generative Adversarial Network (GAN) [29] framework employs two neural networks — a generator $G$ and a discriminator $D$ — that engage in an adversarial training process to synthesize data matching the statistical properties of the training distribution. These networks optimize through a minimax game defined by:

$$\min_{G}\max_{D} V(D, G) = \mathbb{E}_{x \sim p_{data}(x)}[log D(x)] + \mathbb{E}_{z \sim p_z(z)}[log(1 - D(G(z)))], \tag{4}$$

where $p_z(z)$ denotes the noise prior distribution, and $x$ represents samples from the real data distribution $p_{data}$. Through iterative updates, $G$ learns to generate synthetic data indistinguishable from real samples under $D$'s evaluation, thus achieving distributional alignment. For our specialized objective, we use WGAN [45] in this work, the discriminator optimization objective is formulated as:

$$\max_{D} \mathbb{E}_{(\mathbf{s},\mathbf{a},\mathbf{s}') \sim d^{\mathcal{T}}}[D(\mathbf{s}, \mathbf{a}, \mathbf{s}')] - \mathbb{E}_{(\mathbf{s},\mathbf{a},\mathbf{s}') \sim d^{\mathcal{S}}}[D(\mathbf{s}, \mathbf{a}, \mathbf{s}')], \tag{5}$$

where $d^{\mathcal{S}}$ and $d^{\mathcal{T}}$ denote trajectories from the simulator and reality respectively, this objective incentivizes $D$ to discriminate between simulator and reality state-action transitions dynamics. The discriminator used in DEAL adopts a fully connected MLP with input $(\mathbf{s}, \mathbf{a}, \mathbf{s}')$, two hidden layers of 256 units with ReLU activation, and a scalar output. To further stabilize the training process, we use Weight Clipping [45] to located the weights of discriminator in a compact interval to satisfy the Lipschitz continuity condition, ensuring the effective computation of the Wasserstein distance.

## 3.2   DEAL

The schematic overview of the DEAL architecture is shown in Fig. 1, and the pseudo-code is shown in Algorithm 1. Through adversarial co-optimization, this framework progressively minimizes the distribution discrepancies until convergence, achieving a calibrated simulator that becomes indistinguishable from the reality to the discriminator.

**Diffusion Evolution Adversarial Learning**   In this work, by modeling the processes in reality and simulation as MDPs, the system identification objective is to narrow the gap between the state transitions of the target (reality) and source (simulator) domains. Due to different physical dynamic parameters, there's a reducible reality gap between the target domain state transition distribution $\mathcal{P}_t(\theta^{target})$ and the source domain one $\mathcal{P}_s(\theta)$, $\theta$ denotes the physical parameters of the robot or the external environment, as detailed in Appendix A.7. Our method aims to search a parameter distribution $\theta$ which is modeled as a Gaussian distribution to minimize this gap. Specifically, the goal of DEAL is to minimize the discrepancy between state transition distributions from different domains as follows:

$$\min_{\theta \in \mathcal{U}} \|\mathcal{P}_t(\theta^{target}), \mathcal{P}_s(\theta)\|. \tag{6}$$

The similarity of state transitions between parameterized simulations and reality can be measured by a fitness function $f(\cdot)$, which is replaced by a discriminator network in this work. The discriminator evaluates how similar the state transitions in the trajectories are to those in the target domain and gives a corresponding score based on the level of similarity. With DE serves as the generator and combined with Equation (5), the overall search objective can be described as:

$$\theta^* = \arg\min_{\theta \in \mathcal{U}}\max_{D} \mathbb{E}_{(\mathbf{s},\mathbf{a},\mathbf{s}') \sim d^{\mathcal{T}}(\theta^{target}, \pi_0)}[D(\mathbf{s}, \mathbf{a}, \mathbf{s}')] - \mathbb{E}_{(\mathbf{s},\mathbf{a},\mathbf{s}') \sim d^{\mathcal{S}}(\theta, \pi_0)}[D(\mathbf{s}, \mathbf{a}, \mathbf{s}')], \tag{7}$$

where $d^{\mathcal{S}}(\theta, \pi_0)$ represents the trajectory collected by $\pi_0$ in the simulator parameterized by $\theta$. In this adversarial process, DE acts as the generator to generate and optimize the $\theta$, making the simulation trajectory increasingly similar to the real world trajectory $d^{\mathcal{T}}(\theta^{target}, \pi_0)$ from the discriminator's perspective, while the discriminator aims to distinguish between them as much as possible.

In this frame, firstly, we randomly sample physical parameters $\theta_T$ from the initial search range $\mathcal{U}$ to initialize each simulator, then use the policy $\pi_0$ trained via UDR to collect trajectories that reflect the current environment's dynamics. After sampling state-action transition $(\mathbf{s}, \mathbf{a}, \mathbf{s}')$ sequence $b^{\mathcal{T}}$ and $b^{\mathcal{S}}$ from the target and source domain trajectories respectively, then fed them to the discriminator to evaluate each parameterized simulator's fitness, which represent the similarity of state transitions between the evolving simulator and target environment, it is used to update discriminator according to Equation (5) and guide DE for parameter updates. During the parameter update phase, by mapping the fitness through Softmax to obtain a probability distribution, DE can estimates the optimal parameters $\hat{\theta}_0$ by weighting the current population individuals based on the corresponding fitness probabilities and predicts noise $\hat{\epsilon}$ used for directed evolution. Finally, according to Equation (3), the next generation of parameters can be generated by:

$$\theta_{t-1} = \sqrt{\alpha_{t-1}}\hat{\theta}_0 + \sqrt{1 - \alpha_{t-1} - \sigma_t^2}\hat{\epsilon} + \sigma_t\omega. \tag{8}$$

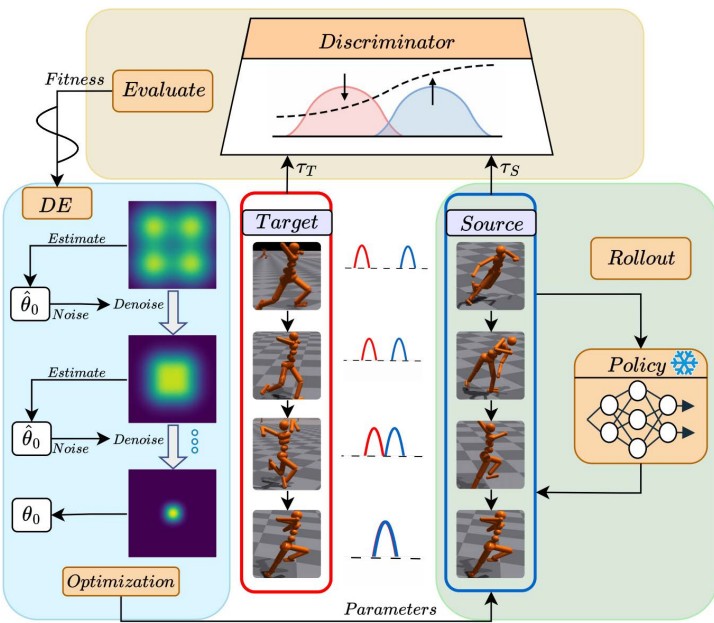

Figure 1: Schematic overview of DEAL. The framework iteratively optimizes parameters through a dual mechanism: a discriminator evaluates the similarity of state transitions sampled in source domain trajectories $\tau_S$ and target domain trajectories $\tau_T$ as fitness, while DE estimates the optimal parameter $\hat{\theta}_0$ based on the fitness probabilities, adaptively updates noise predictions $\hat{\epsilon}$ and performs denoising to optimize parameter distributions until convergence.

**Automatic Noise Adaptaion and Parameter Normalization**    To encourage DEAL to search for the optimum, we use Equation (9) to finetune the noise impact factor $\delta$ base on fitness, then we let $\hat{\sigma}_t = \delta\sigma_t$ to adaptively adjust the noise schedule $\sigma_t$, which can influence the current random perturbation $\omega$ in search process. Specifically, this method adjusts the level of $\omega$ to balance exploration and exploitation, when the parameter population has low fitness, it increases $\omega$ to encourage exploration and generate more mutations in search of better optima. Conversely, when fitness is high, indicating the population is in a good evolutionary state, it reduces $\omega$ to focus on exploitation of the current optimal region.

$$\delta = \lambda \cdot \exp\left(-\mathbb{E}_{(\mathbf{s}, \mathbf{a}, \mathbf{s}') \sim d^{\mathcal{S}}(\mathbf{s}, \mathbf{a}, \mathbf{s}')}[D(\mathbf{s}, \mathbf{a}, \mathbf{s}')]\right) \text{ and } \hat{\sigma}_t = \delta\sigma_t, \tag{9}$$

where $d^{\mathcal{S}}$ represents trajectories collected in source domain and $\lambda$ is the influence coefficient, $D$ is the discriminator quantifies the fitness of each evolving simulator. Combined with Equation (8), the

---

**Algorithm 1:** DEAL

---

**Input:** Population size $N$, Parameter dimension $K$, Search steps $T$, Sample batch size $M$,
   UDR policy $\pi_0$, Search range $\mathcal{U}$, Target domain demonstration $\mathcal{D}_\mathcal{T}$, Source domain
   demonstration $\mathcal{D}_\mathcal{S}$, Diffusion schedule $\alpha_t$, Noise schedule $\sigma_t$, Noise factor $\delta$,
   Random perturbations $\omega$, Probability mapping function $g(\cdot)$.

**Output:** Optimal parameter distribution $\theta^* = \theta^0$

1  Initialize parameters: $\theta^T = [\theta_1^T, \theta_2^T, ..., \theta_N^T] \leftarrow \mathcal{U}$
2  **for** $t \in [T, T-1, ..., 1]$ **do**
3   $\quad \forall j \in [1, N]$: Store $Rollout(\pi_0, Sim(\theta_j^{(t)}))$ in $\mathcal{D}_\mathcal{S}$
4   $\quad$ **for** *update and clip network weights step* $= 0, 1, 2, \cdots, n$ **do**
5   $\quad\quad b^\mathcal{T} = (\mathbf{s}_i, \mathbf{a}_i, \mathbf{s}_i')_{i=1}^M \leftarrow Sample(\mathcal{D}_\mathcal{T}), b^\mathcal{S} = (\mathbf{s}_i, \mathbf{a}_i, \mathbf{s}_i')_{i=1}^M \leftarrow Sample(\mathcal{D}_\mathcal{S})$
6   $\quad\quad$ Update $D$ according to Equation (5) using $b^\mathcal{T}$ and $b^\mathcal{S}$
7   $\quad$ **end**
8   $\quad$ Evaluate fitness and Map to Probability:
9   $\quad \forall j \in [1, N]$: $fitness = \mathbb{E}_{\tau_j \sim Rollout(\pi_0, Sim(\theta_j^{(t)}))}[D(\mathbf{s}, \mathbf{a}, \mathbf{s}')], p_j \leftarrow g(fitness)$
10  $\quad$ Calculate noise factor $\delta$ and Adjust noise schedule $\sigma_t$ according to Equation (9)
11  $\quad$ **for** *env* $j = 1, 2, \cdots, N$ **do**
12  $\quad\quad Z \leftarrow \sum_{i=1}^N p_i \cdot \mathcal{N}(\theta_t^{(j)}; \sqrt{\alpha_t}\theta_t^{(i)}, 1 - \alpha_t)$
13  $\quad\quad \hat{\theta}_0 \leftarrow \frac{1}{Z} \sum_{i=1}^N p_i \cdot \mathcal{N}(\theta_t^{(j)}; \sqrt{\alpha_t}\theta_t^{(i)}, 1 - \alpha_t)\theta_t^{(i)}$
14  $\quad\quad \hat{\epsilon} \leftarrow \frac{\theta_t^{(j)} - \sqrt{\alpha_t}\hat{\theta}_0}{\sqrt{1 - \alpha_t}}, \omega \leftarrow \mathcal{N}(0, I^K)$
15  $\quad\quad \theta_{t-1}^{(j)} \leftarrow \sqrt{\alpha_{t-1}}\hat{\theta}_0 + \sqrt{1 - \alpha_{t-1} - \hat{\sigma}_t^2}\hat{\epsilon} + \hat{\sigma}_t\omega$
16  $\quad$ **end**
17 **end**

---

generation equation can be rewritten as:

$$\theta_{t-1} = \sqrt{\alpha_{t-1}}\hat{\theta}_0 + \sqrt{1 - \alpha_{t-1} - \hat{\sigma}_t^2}\hat{\epsilon} + \hat{\sigma}_t\omega. \tag{10}$$

This mechanism balances exploration and exploitation for DEAL, encouraging thorough parameter space search while converging toward high-fidelity simulations.

Previous learning based Sys-Id methods [14, 16, 18, 22] have crossover errors due to the parameter scale imbalances. For instance, when searching for motor 's stiffness and damping, their values can differ by tens to hundreds of times, such parameters have unbalanced values and the optimization step lengths don't match may lead to crossover errors during system identification. In this work, we normalize the parameters within their initial search range $\mathcal{U}$, this approach allows for a percentage based adjustment of parameters, effectively reducing search errors caused by parameter scale mismatches.

## 4 Experiments

In this section, we compare DEAL with several baselines across multiple tasks to demonstrate its effectiveness. The key aspects are as follows:

1. We search for parameters in multiple tasks, including high-dimensional scenarios, then compare the search errors of DEAL and other baselines.

2. After enhancing the simulator across various tasks, we evaluate the transfer performance of the retrained policies using DEAL and other baselines.

3. We search for parameters with larger search scales to examine the search adaptability and assess the demonstration requirements of DEAL and other baselines.

4. Finally, we complete the challenging sim-to-real transfer and demonstrate the performance improvements brought by DEAL.

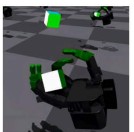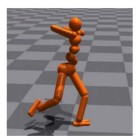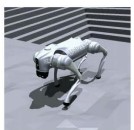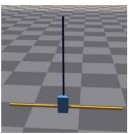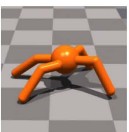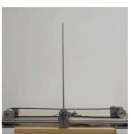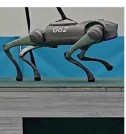

Figure 2: Experiment tasks in simulation (1∼5): AllegroHand, Humanoid, Go2, Cartpole and Ant. Experiment tasks in reality (6∼7): Cartpole, Go2. Presented in order from left to right.

**Task Setup**    We test DEAL in 5 sim-to-sim tasks (AllegroHand, Humanoid, Go2, Cartpole, Ant) and 2 sim-to-real task (Cartpole, Go2), all environments are shown in th Fig. 2. **(1)AllegroHand** environment is a AllegroHand robot designed to interact with a cube, it features 16 DoF distributed across the hand and fingers. The task is to use the hand to manipulate an object to align its orientation with the target orientation. **(2) Humanoid** environment contain a humanoid robot has 21 DoF, with 3 for the abdomen, 6 each for the right and left legs, and 3 each for the right and left arms. Its tasks include walking stably and performing various movements to achieve specific speeds/directions. **(3) Go2** environment is a 12-DoF Unitree GO2 quadruped, with each leg having 3-DoF. The task is to follow randomly chosen x, y, yaw target velocities and climb platforms. **(4) Cartpole** environment includes an inverse pendulum connected to a 1-DoF cart. The task is to keep the pendulum on the cart balanced for as long as possible. **(5) Ant** environment is an 8-DOF quadrupedal robot consisting of four legs attached to a common base. The task is controlling the ant run as fast as possible.

**Experiment Detail**    Our experiments employed Isaac Gym [46] as the simulator. In Isaac Gym, we can parallelly collect trajectories in hundreds of environments with different parameters which means we could evaluate hundreds of parameters parallelly. During parameter search, we instantiat 200 parallel environments to assess reality-to-simulation alignment across varying physical parameters. For Cartpole, Ant, Humanoid and AllegroHand, we implement Soft Actor-Critic (SAC) [47] to train a neural network as RL controller. For Go2, we adopt RMA [33] to develop controllers tracking velocity commands and climbing platforms, and train canter controllers using Ess-InfoGAIL [48] for bio-inspired running and command following. In our experiment, running on a PC equipped with Intel i5-14600KF and RTX 4060 Ti, DEAL can complete the entire search process within a few minutes, the search computation cost can be found in the Appendix A.2. Additionally, we provide the simulation data budgets for each task, the detailed numbers are shown in Appendix A.3.

**Baselines**    We design various baselines to demonstrate the efficiency of our method. **(1) EASI [14]**: A learning-based Sys-Id method combines CMA-ES [49] and GAN [29]. **(2) Bayes Optimization (BO)**: Modification of Bayesian Sys-Id approach—BayRn [22] in search tasks with its return evaluated by a discriminator. **(3) DEAL\FN**: DEAL without using Automatic Noise Adaptation base on fitness **(4) DEAL\PN**: DEAL without using Parameter Normalization **(5) Uniform Domain Randomization(UDR)**: Uniformly sampling parameters from $\mathcal{U}$ at the beginning of each training iteration. **(6) Oracle**: Direct training in the target environment in sim-to-sim experiments, representing the ideal upper bound for sim-to-sim transfer tasks.

### 4.1   Parameter Search Capability

In this experiment, we selected five tasks and conducted a thorough parameter search for each. The parameter list is detailed in the Appendix A.7 and mainly covers friction coefficients, restitution coefficients, masses of rigid bodies and motor properties such as stiffness, damping and friction. When the parameters of target domain are unknown, we typically train a policy across a wide range of simulation parameters in hope that the policy can thus handle possible real-world variations in dynamics or observations. In this experiment, the UDR policy $\pi_0$ of Ant and Go2 was trained within the range of $[1/3 \times \theta_t, 3 \times \theta_t]$, the remain tasks were trained within the range of $[1/5 \times \theta_t, 5 \times \theta_t]$, where $\theta_t$ denoting the parameters of the target domain. The initial search range $\mathcal{U}$ for parameters of each task is the same as its training range, we performed 50 steps of parameter search for each task with limited 'real-world' demonstration collected by $\pi_0$. To avoid randomness, we presented the experimental outcomes using the average values from multiple random seeds. The evaluation metric is the average percentage error of the searched parameters relative to the true parameters, defined as follows:

$$\epsilon_p = \mathbb{E}\left(\left|\frac{\theta - \theta^{target}}{\theta^{target}}\right|\right). \tag{11}$$

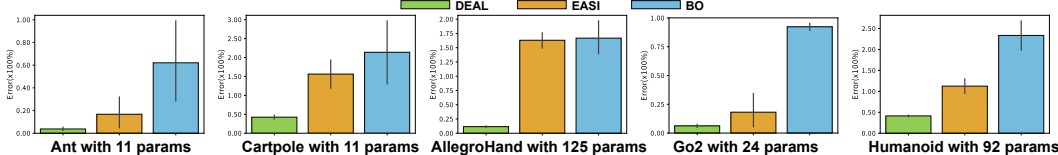

Figure 3: Average search errors for each method, consisting of the error percentage of all parameters.

As shown in Fig. 3, when comparing the final search results of the three methods with the target parameter errors, it is evident that DEAL exhibits significantly higher search accuracy than other baselines across all environments, especially in high-dimensional environments. The baselines often suffer from high-dimensional parameter identification collapse and fail to converge to precise solutions. Despite using less computation cost and the same demonstration in this experiment, DEAL achieve optimal performance which demonstrates its powerful search capability and stable performance.

## 4.2 Sim-to-Sim Transfer

In this experiment, we utilize the simulator updated from the previous experiment for new policy training, then transfer these newly trained policies to target environment to compare the performance with UDR and Oracle. Notably, the target environment in this experiment is the simulator set with the true target parameter. This is done to compare the performance improvements brought by various enhanced simulators to the policy. As shown in Fig. 4 left, the simulator improved by DEAL delivers the greatest performance boost to newly trained policies for all tasks, it is close to or only slightly inferior to the performance of Oracle whose policy is trained directly in the target domain. Moreover, in high dimensional tasks like Humanoid and AllegroHand, due to the high-dimensional parameter identification collapse of the baselines, the simulator undergoes negative optimization, the policies trained in such simulator perform worse than those trained by UDR and may even cause training to crash, fail to converge, and lead to abnormal behavior. For Go2 task, after training an initial policy

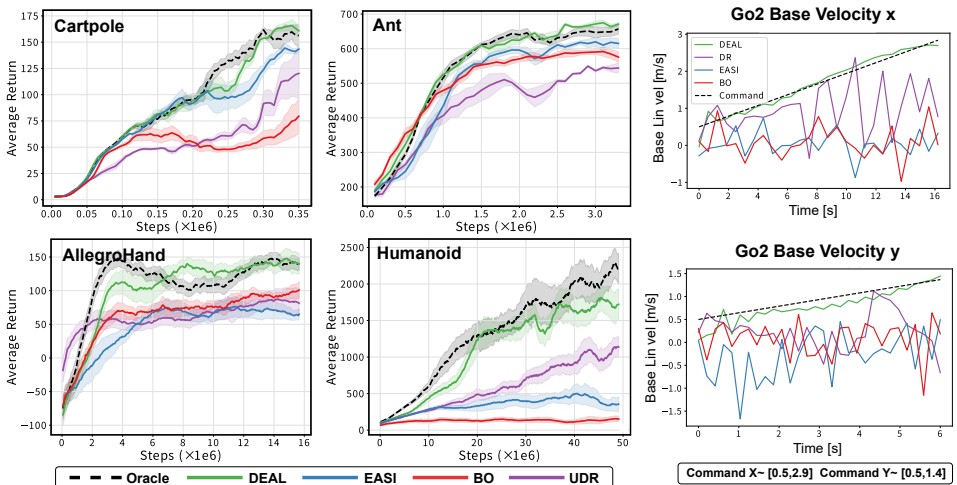

Figure 4: **Left**: The average return in the target environment for the policies trained at each stage with DEAL and other sim-to-real baselines. **Right**: The speed tracking display of Go2 under training with DEAL and other sim-to-real baselines. up: speed in the robot's x-axis direction, down: speed in the robot's y-axis direction.

to collect target-domain trajectories, we retrain the policy in the simulator optimized by DEAL and other baselines. Then, we compare the adaptability of the final policy in target environment, this experiment examines the speed-tracking capability of policies, whether the controller can control the Go2 robot to move forward in the specified direction and speed, and to ascend steps. As shown

in Fig. 4 right, DEAL achieves the best performance as the robot's movement accurately tracks the speed commands in both forward and vertical directions. However, the policies derived from other baselines fail to sustain tracking of velocity commands when they are progressively increased, ultimately leading to more severe speed collapse and destabilizing oscillations in robot motion than UDR. This experiment demonstrates DEAL significantly enhances transfer performance across all tasks.

### 4.3 Parameter Search Adaptability and Data Requirements

Table 1: Average search error percentage ($\times 100\%$) (See Appendix A.8 for error bars).

| | Cartpole | | | Humanoid | | | AllegroHand | | |
|---|---|---|---|---|---|---|---|---|---|
| Method | $\xi =10$ | $\xi =15$ | $\xi =20$ | $\xi =10$ | $\xi =15$ | $\xi =20$ | $\xi =10$ | $\xi =15$ | $\xi =20$ |
| DEAL | **0.85** | **1.74** | **2.63** | **0.81** | **1.65** | **2.50** | **0.83** | **1.69** | **2.57** |
| DEAL\PN | 1.90 | 2.85 | 3.76 | 2.40 | 3.53 | 4.58 | 2.60 | 3.80 | 4.92 |
| DEAL\FN | 1.67 | 2.96 | 4.28 | 1.57 | 2.79 | 4.03 | 1.59 | 2.80 | 4.06 |
| EASI | 4.03 | 6.12 | 8.40 | 2.94 | 4.64 | 6.03 | 4.25 | 6.82 | 9.32 |
| BO | 5.20 | 8.52 | 11.78 | 4.67 | 7.29 | 10.06 | 3.75 | 6.32 | 9.16 |

In this experiment, we search for parameters on larger search scales and determine the requirements for demonstration of DEAL and other baselines. As shown in Table 1, denoting $\theta_t$ as the target parameters, DEAL attain the minimal search errors in the initial search ranges of $[\frac{1}{\xi} \times \theta_t, \xi \times \theta_t]$ across all tasks, and maintain estimation accuracy even under severely limited parameter distribution priors and extreme initial values during large-scale searches. Notably, UDR policies trained with a range of $[\frac{1}{10} \times \theta_t, 10 \times \theta_t]$ have started to fail in the target domain, producing low quality even all failed trajectories that cause other search baselines' errors to surge. Such scenarios are prevalent in real world deployment as the initial UDR policies rarely perform well at the outset. This highlights DEAL's strong search adaptability and stable performance. Furthermore, due to the lack of parallel data collection in reality, the scarcity of data and the high cost of collection mean few shot methods must minimize data requirements. Fig. 5 reveals DEAL's superior data efficiency: While existing Sys-Id methods suffer significant performance degradation with sparse trajectory samples, DEAL sustains robust identification accuracy under data-scarce conditions.

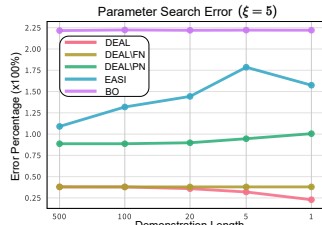

Figure 5: Average search errors of DEAL and other baselines when given different demonstration lengths in Humanoid task.

### 4.4 Sim-to-Real Transfer

In this experiment, we first deploy the UDR-trained policy to collect real-world trajectories in a few seconds, then optimize the simulator with DEAL, retrain the policy and redeploy to compare sim-to-real performance. In the Cartpole task, the controller outputs a target cart position to maintain the pole in balance. Stability is measured by pole-angle errors and cart-velocity, both of which are reduced after DEAL optimization as shown in Table 2, demonstrating a improved sim-to-real performance.

In the Go2 environment, the controller outputs target position signals for each joint at a frequency of 50 Hz. These signals are then converted into motor torques by the motor's built-in PD controller operating at 1000 Hz and applied to the corresponding joints. In this section, we train a climbing policy based

Table 2: Cartpole sim-to-real performance.

| Method | Angle Error $\times 10^{-2}$ | Cart Vel $\times 10^{-1}$ |
|---|---|---|
| UDR | 3.655$\pm$1.122 | 1.480$\pm$0.367 |
| DEAL | 1.372$\pm$0.382 | 1.214$\pm$0.118 |

on RMA [33] for the experiment of climbing onto a high platform, and train a canter policy using Ess-InfoGAIL [48] for the quadruped robot's running experiments. Firstly, we conduct repetitive high-platform climbing experiments to evaluate the policy performance. As shown in Fig. 6, both UDR and Unitree's built-in RL policies are unable to reproduce the climbing behavior trained in simulation, always fail to control Go2 to climb the platform. After optimized by DEAL, the Go2 robot can fluently and successfully complete the task, the probability of success has been significantly

improved. Subsequently, we deploy the canter policy and compare the running performance of the controllers trained with UDR and DEAL. As shown in Fig. 7, the UDR policy fails to reproduce the straight-running gait observed in simulation, resulting in significant deviations in both the magnitude and direction of the velocity. In contrast, the policy retrained with DEAL accurately tracks the velocity commands and maintains a forward speed close to that achieved in simulation during straight-line locomotion. These results indicate DEAL indeed narrows the sim-to-real gap and enhances the performance of the retrained policy in reality.

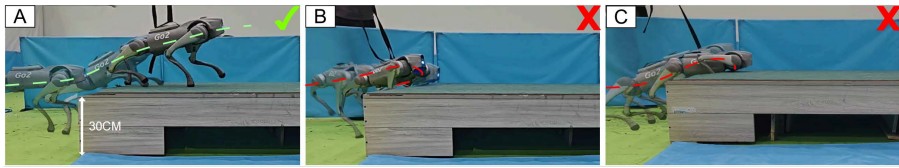

Figure 6: **(A)** Deploy with the climbing controller after being optimized with DEAL. **(B)** Deploy with the climbing controller trained with UDR. **(C)** Deploy with Unitree's built-in RL controller.

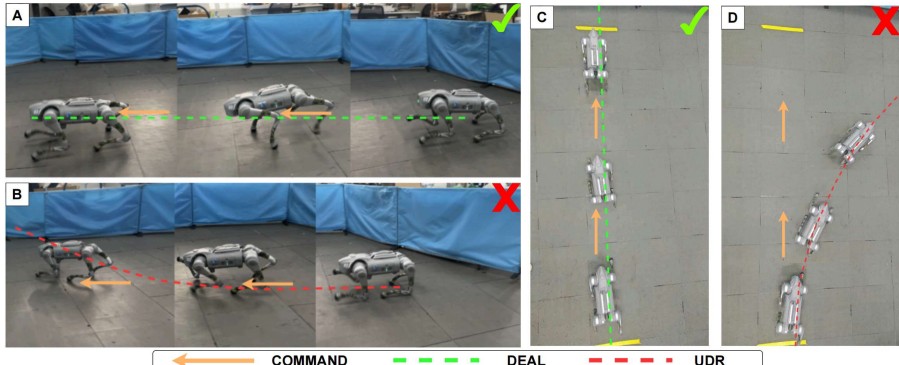

Figure 7: **(A, C)** Deploy with the canter controller after being optimized with DEAL. A: side-view imaging, C: top-view imaging. **(B, D)** Deploy with the canter controller trained with UDR. B: side-view imaging, D: top-view imaging.

## 5 Limitations

Although DEAL demonstrates remarkable performance in high-dimensional parameter optimization tasks under limited demonstration data, its applicability may face challenges in environments with diverse state transition distributions. This potential limitation arises from the inherent binary classification nature of the GAN discriminator, which is designed to distinguish between one target environment transition and one simulator transition at a time. As a result, the discriminator may struggle to provide multimodal fitness evaluations during optimization, suggesting an area for future improvement in handling more complex or multimodal environmental dynamics.

## 6 Conclusion

In this work, we propose DEAL, a novel Sys-Id framework that combines diffusion evolution with adversarial learning for sim-to-real transfer. DEAL optimizes simulator parameters through a dual mechanism: a discriminator evaluates the similarity of state transitions between evolving simulators and reality as fitness guidance, while DE adaptively refines parameter distributions to narrow the reality gap by using fitness-driven denoising. Extensive experiments in simulation and real-world demonstrate DEAL's superior performance in high-dimensional parameter identification and robust transfer performance across challenging tasks, outperforming baselines with minimal real-world data and lower computation cost. We believe that DEAL advances sim-to-real transfer and offers a promising approach for deploying RL controllers in the real world.

## Acknowledgments and Disclosure of Funding

This work was supported in part by the Major Science and Technology Project of Jiangsu Province under Grant BG2024041, National Key Research and Development Program of China under Grant 2023YFD2001003, and the Fundamental Research Funds for the Central Universities under Grant 011814380048.

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

# A Appendix

## A.1 Proof of Diffusion Evolution

In the diffusion phase, let $x_0$ denote the original data point and $x_T$ becomes entirely Gaussian noise. This process can be expressed as:

$$x_t = \sqrt{\alpha_t}x_0 + \sqrt{1 - \alpha_t}\epsilon, \tag{12}$$

where the total noise $\epsilon \sim \mathcal{N}(0, I)$ added to the data $x_0$ at time step $t \in [0, T]$ is controlled by $\alpha_t$, which decreases $\alpha_0 = 1$ to $\alpha_T \approx 0$. Meanwhile, a noise prediction module $\epsilon_\theta$ is trained to minimize this loss function in diffusion phase:

$$\mathcal{L} = \mathbb{E}_{x_0 \sim p_{data}, \epsilon \sim \mathcal{N}(0,I)} \left\| \epsilon_\theta \left( \sqrt{\alpha_t}x_0 + \sqrt{1 - \alpha_t}\epsilon, t \right) - \epsilon \right\|^2, \tag{13}$$

where $p_{data}$ denotes the distribution of the training data. Under the DDIM framework [25], the denoising phase is defined as:

$$x_{t-1} = \sqrt{\alpha_{t-1}} \left( \frac{x_t - \sqrt{1 - \alpha_t}\epsilon_\theta(x_t, t)}{\sqrt{\alpha_t}} \right) + \sqrt{1 - \alpha_{t-1} - \sigma_t^2}\epsilon_\theta(x_t, t) + \sigma_t\omega, \tag{14}$$

where $\sigma_t$ controls the amount of random perturbations $\omega \sim \mathcal{N}(0, I)$ added during the denoising phase. Given the diffusion process by Equation (12), $x_0$ can be expressed by the noise $\epsilon$, and vise versa:

$$x_0 = \frac{x_t - \sqrt{1 - \alpha_t}\epsilon}{\sqrt{\alpha_t}}, \text{ and } \epsilon = \frac{x_t - \sqrt{\alpha_t}x_0}{\sqrt{1 - \alpha_t}}, \tag{15}$$

by substituting Equation (15) to Equation (14) to estimate $\hat{x}_0$ and $\hat{\epsilon}$, the denosing phase can be rewritten as:

$$x_{t-1} = \sqrt{\alpha_{t-1}}\hat{x}_0 + \sqrt{1 - \alpha_{t-1} - \sigma_t^2}\hat{\epsilon} + \sigma_t\omega. \tag{16}$$

Since the denoising step in diffusion models requires an estimation of $x_0$, DE derive it from sample $x$ and the corresponding fitness $f(x)$. The estimation of $x_0$ can be expressed as a conditional probability $p(x_0 = x | x_t)$. Using Bayes' theorem and $p(x_0 = x) = g[f(x)]$ yields:

$$p(x_0 = x | x_t) = \frac{p(x_t | x_0 = x)p(x_0 = x)}{p(x_t)} = \frac{p(x_t | x)g[f(x)]}{p(x_t)}. \tag{17}$$

Here, $p(x_t | x_0 = x)$ can be computed easily by $\mathcal{N}(x_t; \sqrt{\alpha_t}x, 1 - \alpha_t)$ given the design of the diffusion process. Since deep-learning-based diffusion models are trained using mean squared error loss, the $x_0$ estimated by $x_t$ should be the weighted average of the sample $x$. Hence, the estimation function of $x_0$ becomes:

$$\hat{x}_0(x_t, \alpha, t) = \sum_{x \sim p_{\text{eval}}(x)} p(x_0 = x | x_t)x = \sum_{x \sim p_{\text{eval}}(x)} g[f(x)]\frac{\mathcal{N}(x_t; \sqrt{\alpha_t}x, 1 - \alpha_t)}{p(x_t)}x, \tag{18}$$

where $p_{\text{eval}}$ is the evaluation sample on which we compute the fitness score, here given by the current population $X_t = (x_t^{(1)}, x_t^{(2)}, ..., x_t^{(N)})$ of $N$ individuals. Equation (18) has three weight terms: The first term $g[f(x)]$ assigns larger weights to high fitness samples. For each individual sample $x_t$, the second Gaussian term $\mathcal{N}(x_t; \sqrt{\alpha_t}x, 1 - \alpha_t)$ makes each individual only sensitive to local neighbors of evaluation samples. The third term $p(x_t)$ is a normalization term. Hence, $\hat{x}_0$ can be simplified to:

$$\hat{x}_0(x_t, \alpha, t) = \frac{1}{Z} \sum_{x \in X_t} g[f(x)]\mathcal{N}(x_t; \sqrt{\alpha_t}x, 1 - \alpha_t)x, \tag{19}$$

where $Z$ is the normalization term:

$$Z = p(x_t) = \sum_{x \in X_t} g[f(x)]\mathcal{N}(x_t; \sqrt{\alpha_t}x, 1 - \alpha_t). \tag{20}$$

When substituting Equation (19) into Equation (15) we can express noise prediction $\hat{\epsilon}$ as:

$$\hat{\epsilon}(x_t, \alpha, t) = \frac{x_t - \sqrt{\alpha_t}\,\hat{x}_0(x_t, \alpha, t)}{\sqrt{1 - \alpha_t}}. \tag{21}$$

After we successfully estimate the optimal point $\hat{x}_0$ and the noise $\hat{\epsilon}$, the next generation of parameters can be generated by using Equation (16).

Previous black-box optimization methods such as evolution strategies (ES) typically model the parameter distribution as a Gaussian, updating its mean and covariance in each sample iteration. However, the new samples are randomly drawn from the updated distribution and lack explicit control over the convergence trajectory, often resulting in slow convergence and low final accuracy. Instead of blindly perturbing parameters in each iteration, DE performs a re-weighted posterior estimation of the optimum based on observed samples. This allows the search process to focus more quickly on regions close to real data distribution.

Specifically, DEAL updates the parameter distribution using the Equation (10), where the estimated optimal parameter $\hat{\theta}_0$ is given by Equation (1), which can be rewritten as follows:

$$\hat{\theta}_0(\theta_t, \alpha, t) = \frac{1}{Z} \sum_{\theta \in \Theta_t} \underbrace{g[f(\theta)]}_{p(\theta_0=\theta)} \underbrace{\mathcal{N}(\theta_t; \sqrt{\alpha_t}\theta, 1 - \alpha_t)}_{p(\theta_t|\theta_0=\theta)} \theta, \quad g(x_i) = \frac{e^{x_i}}{\sum_j e^{x_j}}. \tag{22}$$

In Equation (22), $p(\theta_t \mid \theta_0 = \theta)$ acts as a neighborhood weight around parameter $\theta_t$. $p(\theta_0 = \theta)$ represents a probability derived by applying a softmax function $g(\cdot)$ to the discriminator's output, reflecting the likelihood that a state transition under parameter $\theta$ resembles one from the real world. This formulation effectively uses a softmax-style weighting to emphasize high-fitness regions, guiding $\hat{\theta}_0$ rapidly toward the true optimal parameters. As a result, the estimate $\hat{\theta}_0$ serves as a proxy for the gradient direction of the WGAN's objective function $\mathcal{L}$ in Equation (23) as follows:

$$\underbrace{\mathcal{L}}_{\searrow} = \mathbb{E}_{(\mathbf{s},\mathbf{a},\mathbf{s}')\sim d^{\mathcal{T}}(\theta^{target}, \pi_0)}[D(\mathbf{s}, \mathbf{a}, \mathbf{s}')] - \underbrace{\mathbb{E}_{(\mathbf{s},\mathbf{a},\mathbf{s}')\sim d^{\mathcal{S}}(\theta, \pi_0)}[D(\mathbf{s}, \mathbf{a}, \mathbf{s}')]}_{\nearrow}. \tag{23}$$

The diffusion noise schedule also helps to stablize the convergence by determining the update step at each iteration. As $t \to 0$, $\alpha_t \to 1$ and $\sigma_t \to 0$, reducing the variance of $\mathcal{N}(\theta_t; \sqrt{\alpha_t}\theta, 1 - \alpha_t)$ and encouraging fine-grained, local updates in the later stages. The update $\theta_{t-1} \to \hat{\theta}_0$ serves as a denoising step, similar to the reverse process in diffusion models. With a well-trained discriminator, this process enables stable convergence toward the true optimum.

Additionally, unlike traditional ES that rely on elitism or truncation, which can lose global information and lead to premature convergence to local optima, DEAL utilizes the entire population in Equation (22). It assigns soft weights based on discriminator-derived fitness to estimate the gradient direction of the WGAN's objective function. This leads to more targeted and stable updates, offering improved convergence stability, efficiency and accuracy over previous methods.

### A.2 Computation cost

The computation time cost presented in Table 3 is based on searches conducted over 50 steps on a PC equipped with an Intel i5-14600KF and an RTX 4060 Ti.

Table 3: Average computation time cost.

| Task | DEAL | EASI | BO |
|------|------|------|------|
| Ant | **1 min 38 s** | 1 min 43 s | 7 min 05 s |
| Cartpole | **1 min 15 s** | 1 min 35 s | 6 min 55 s |
| Go2 | **5 min 03 s** | 5 min 26 s | 21 min 17 s |
| Humanoid | **1 min 43 s** | 1 min 52 s | 42 min 54 s |
| AllegroHand | **54 min 05 s** | 54 min 15 s | 125 min 04 s |

### A.3 Data budget

All tasks run 200 parallel environments, but the trajectory lengths differ across tasks. Table 4 below lists the data budget allocated to DEAL and the baseline methods for each task.

Table 4: Data Budget (200 parallel environments $\times$ trajectory length).

| Data budget | Ant | CartPole | Humanoid | Go2 | AllegroHand |
|---|---|---|---|---|---|
| For DEAL and baselines | $200 \times 200$ | $200 \times 200$ | $200 \times 200$ | $200 \times 250$ | $200 \times 600$ |

## A.4 Impact of trajectories quality

Current learning-based system identification methods do require the collection of trajectories with a certain quality in the real world. However, even trajectories from very poor policies still encode physical dynamics information. When using the same policy to search parameters, the failed behavior should be reproduced by adjusting those parameters. To address the concern regarding DEAL's reliance on trajectory quality, we train the UDR policies for the CartPole and Humanoid in simulation, and take the policies at various training checkpoints to collect a 200-step trajectory in the target environment. We then search for parameters within $[\frac{1}{3} \times \theta^{target}, 3 \times \theta^{target}]$ to reflect the impact of the policies performance on the search results. In the Tables 5, 50 denotes the policy after 50 training iterations, and the policy only starts to converge in the last row.

Table 5: Average search results at each checkpoint.

| Iterations(CartPole) | Avg. Search Error (%) | Iterations(Humanoid) | Avg. Search Error (%) |
|---|---|---|---|
| 50 | $8.84 \pm 2.96$ | 5e3 | $13.59 \pm 5.37$ |
| 100 | $7.87 \pm 2.35$ | 1e4 | $14.70 \pm 4.80$ |
| 250 | $7.19 \pm 2.96$ | 2e4 | $13.08 \pm 5.00$ |
| 1000 | $7.34 \pm 2.88$ | 2.5e4 | $10.24 \pm 4.13$ |
| 2500 | $5.10 \pm 2.02$ | 3e4 | $10.78 \pm 3.40$ |

Contrary to concerns about its reliance on trajectory quality, DEAL demonstrates strong robustness and can still accurately search parameters even when using low-quality trajectories collected from early-stage policies. As training progresses and trajectory quality improves, DEAL's search performance improves accordingly.

## A.5 Comparison with model-based methods

To assess the improvement of DEAL in finding the optimal parameters in comparison with a non-convex optimization problem formulation. We further compare DEAL with several strong model-based baselines in the case of a simple robotic system with non-linear dynamics, including least-squares estimation and Extended Kalman Filter(EKF). In this section, we conduct parameter search experiments on the CartPole task using the same trajectory data for all three methods, the results are reported in the Table 6.

Table 6: Comparison with model-based methods.

| Avg. Search Error (%) | CartPole |
|---|---|
| DEAL | **6.9$\pm$1.1** |
| Model-based EKF | 37.4$\pm$8.8 |
| Least-squares | 53.0$\pm$19.6 |

As shown in the table, EKF suffers from significant bias due to linearization errors when estimating nonlinear parameters. The least-squares estimation exhibits relatively large search errors, and the results are highly sensitive to random initialization. In contrast, DEAL demonstrates stable and accurate search performance in such nonlinear systems.

## A.6 Broader impacts

In our work, we propose DEAL, a novel framework that leverages diffusion evolution and adversarial learning to align simulated and real-world dynamics with minimal real-world demonstration. By

narrowing the reality gap, DEAL can accelerate the deployment of RL controllers in robotics, reducing the need for costly and time-consuming hardware trials. This has the potential to democratize advanced robotic applications across industries—manufacturing, logistics, healthcare by lowering both development time and resource requirements. At the same time, overreliance on simulator-guided optimization carries the risk that unmodeled real-world complexities—sensor degradation, unanticipated contacts may still lead to failures during long-term operation. Thus, while DEAL substantially improves sim-to-real transfer, it should be complemented by continued investment in robust robot design, sensor redundancy, and online adaptation schemes. Future work should explore hardware–software co-design to further to address the sim-to-real challenge.

## A.7 Physical parameter list

The parameters that are symmetrically distributed on the robot's limbs should be chosen to be the same or similar values to ensure successful training, however, during the parameter search process, they are still treated as independent parameters for searching.

Table 7: Parameter list in the Ant Environment.

| Parameter | Target Value | Unit |
|---|---|---|
| Contact Friction | 1.5 | - |
| Contact Restitution | 0.01 | - |
| Body-Leg Motor Friction | 0.2 | - |
| Body-Leg Motor Damping | 0.3 | $N \cdot m \cdot s/rad$ |
| Body-Leg Motor Armature | 0.1 | $kg \cdot m^2$ |
| Foot-Leg Motor Friction | 0.1 | - |
| Foot-Leg Motor Damping | 0.2 | $N \cdot m \cdot s/rad$ |
| Foot-Leg Motor Armature | 0.1 | $kg \cdot m^2$ |
| Foot Mass | 0.1 | kg |
| Leg Mass | 0.2 | kg |
| Body Mass | 1.0 | kg |

Table 8: Parameter list in the Cartpole Environment.

| Parameter | Target Value | Unit |
|---|---|---|
| Pole Length | 0.3 | m |
| Pole Mass | 0.1 | kg |
| Cart Mass | 0.3 | kg |
| Pole DOF_Friction | 0.1 | - |
| Pole DOF_Damping | 0.1 | $N \cdot m \cdot s/rad$ |
| Pole DOF_Amature | 0.2 | $kg \cdot m^2$ |
| Cart DOF_Friction | 0.1 | - |
| Cart PID_P | 0.1 | $N \cdot m/rad$ |
| Cart PID_D | 0.1 | $N \cdot m \cdot s/rad$ |
| Cart EffortLimit | 0.2 | N |
| Cart Vel | 1.0 | m/s |

Table 9: Parameter list in the Go2 Environment.

| Parameter | Target Value | Unit |
|---|---|---|
| Hip Damping$\times 4$ | 20 | $N \cdot m \cdot s/rad$ |
| Hip Stiffness$\times 4$ | 0.5 | $N \cdot m/rad$ |
| Calf Damping$\times 4$ | 20 | $N \cdot m \cdot s/rad$ |
| Calf Stiffness$\times 4$ | 0.5 | $N \cdot m/rad$ |
| Thigh Damping$\times 4$ | 20 | $N \cdot m \cdot s/rad$ |
| Thigh Stiffness$\times 4$ | 0.5 | $N \cdot m/rad$ |

Table 10: Parameter list in the AllegroHand Environment.

| Parameter | Target Value | Unit |
|---|---|---|
| Contact Friction×22 | 1.0 | - |
| Contact Restitution×22 | 0.01 | - |
| Dof Friction×16 | 0.01 | - |
| Dof Damping×16 | 0.1 | $N \cdot m \cdot s/rad$ |
| Dof Armature×16 | 0.001 | $kg \cdot m^2$ |
| Dof Stiffness×16 | 3.0 | $N \cdot m/rad$ |
| Allegro_Mount Mass | 0.47 | kg |
| Index_Link_0 Mass | 0.012 | kg |
| Index_Link_1 Mass | 0.065 | kg |
| Index_Link_2 Mass | 0.036 | kg |
| Index_Link_3 Mass | 0.031 | kg |
| Middle_Link_0 Mass | 0.012 | kg |
| Middle_Link_1 Mass | 0.065 | kg |
| Middle_Link_2 Mass | 0.036 | kg |
| Middle_Link_3 Mass | 0.031 | kg |
| Ring_Link_0 Mass | 0.012 | kg |
| Ring_Link_1 Mass | 0.065 | kg |
| Ring_Link_2 Mass | 0.036 | kg |
| Ring_Link_3 Mass | 0.031 | kg |
| Thumb_Link_0 Mass | 0.018 | kg |
| Thumb_Link_1 Mass | 0.012 | kg |
| Thumb_Link_2 Mass | 0.038 | kg |
| Thumb_Link_3 Mass | 0.060 | kg |

Table 11: Parameter list in the Humanoid Environment.

| Parameter | Target Value | Unit |
|---|---|---|
| Contact Friction×4 | 1.0 | - |
| Contact Restitution×4 | 0.01 | - |
| Abdomen_Y Friction | 0.01 | - |
| Abdomen_Y Damping | 5.0 | $N \cdot m \cdot s/rad$ |
| Abdomen_Y Armature | 0.02 | $kg \cdot m^2$ |
| Abdomen_Y Stiffness | 20.0 | $N \cdot m/rad$ |
| Abdomen_Z Friction | 0.01 | - |
| Abdomen_Z Damping | 5.0 | $N \cdot m \cdot s/rad$ |
| Abdomen_Z Armature | 0.01 | $kg \cdot m^2$ |
| Abdomen_Z Stiffness | 20.0 | $N \cdot m/rad$ |
| Abdomen_X Friction | 0.01 | - |
| Abdomen_X Damping | 5.0 | $N \cdot m \cdot s/rad$ |
| Abdomen_X Armature | 0.01 | $kg \cdot m^2$ |
| Abdomen_X Stiffness | 10.0 | $N \cdot m/rad$ |
| Hip_X Friction×2 | 0.01 | - |
| Hip_X Damping×2 | 5.0 | $N \cdot m \cdot s/rad$ |
| Hip_X Armature×2 | 0.01 | $kg \cdot m^2$ |
| Hip_X Stiffness×2 | 10.0 | $N \cdot m/rad$ |
| Hip_Z Friction×2 | 0.01 | - |
| Hip_Z Damping×2 | 5.0 | $N \cdot m \cdot s/rad$ |
| Hip_Z Armature×2 | 0.01 | $kg \cdot m^2$ |
| Hip_Z Stiffness×2 | 10.0 | $N \cdot m/rad$ |
| Hip_Y Friction×2 | 0.01 | - |
| Hip_Y Damping×2 | 5.0 | $N \cdot m \cdot s/rad$ |
| Hip_Y Armature×2 | 0.01 | $kg \cdot m^2$ |
| Hip_Y Stiffness×2 | 20.0 | $N \cdot m/rad$ |
| Knee Friction×2 | 0.01 | - |
| Knee Damping×2 | 0.1 | $N \cdot m \cdot s/rad$ |
| Knee Armature×2 | 0.007 | $kg \cdot m^2$ |
| Knee Stiffness×2 | 5.0 | $N \cdot m/rad$ |
| Ankle_X Friction×2 | 0.01 | - |
| Ankle_X Damping×2 | 1.0 | $N \cdot m \cdot s/rad$ |
| Ankle_X Armature×2 | 0.006 | $kg \cdot m^2$ |
| Ankle_X Stiffness×2 | 2.0 | $N \cdot m/rad$ |
| Ankle_Y Friction×2 | 0.01 | - |
| Ankle_Y Damping×2 | 1.0 | $N \cdot m \cdot s/rad$ |
| Ankle_Y Armature×2 | 0.006 | $kg \cdot m^2$ |
| Ankle_Y Stiffness×2 | 2.0 | $N \cdot m/rad$ |
| Shoulder1 Friction×2 | 0.01 | - |
| Shoulder1 Damping×2 | 5.0 | $N \cdot m \cdot s/rad$ |
| Shoulder1 Armature×2 | 0.01 | $kg \cdot m^2$ |
| Shoulder1 Stiffness×2 | 10.0 | $N \cdot m/rad$ |
| Shoulder2 Friction×2 | 0.01 | - |
| Shoulder2 Damping×2 | 5.0 | $N \cdot m \cdot s/rad$ |
| Shoulder2 Armature×2 | 0.01 | $kg \cdot m^2$ |
| Shoulder2 Stiffness×2 | 10.0 | $N \cdot m/rad$ |
| Elbow Friction×2 | 0.01 | - |
| Elbow Damping×2 | 1.0 | $N \cdot m \cdot s/rad$ |
| Elbow Armature×2 | 0.006 | $kg \cdot m^2$ |
| Elbow Stiffness×2 | 2.0 | $N \cdot m/rad$ |

## A.8 Error bars

These error bars are based on calculations from $95\% \, CI$.

Table 12: Average search error percentage($\times 100\%$).

| | Cartpole | | |
|---|---|---|---|
| Method | $\xi =10$ | $\xi =15$ | $\xi =20$ |
| DEAL | **0.85±0.06** | **1.74±0.10** | **2.63±0.13** |
| DEAL\PN | 1.90±0.11 | 2.85±0.15 | 3.76±0.17 |
| DEAL\FN | 1.67±0.10 | 2.96±0.06 | 4.28±0.12 |
| EASI | 4.03±0.50 | 6.12±0.35 | 8.40±0.46 |
| BO | 5.20±0.04 | 8.52±0.16 | 11.78±0.25 |

Table 13: Average search error percentage($\times 100\%$).

| | Humanoid | | |
|---|---|---|---|
| Method | $\xi =10$ | $\xi =15$ | $\xi =20$ |
| DEAL | **0.81±0.02** | **1.65±0.04** | **2.50±0.05** |
| DEAL\PN | 2.40±0.06 | 3.53±0.07 | 4.58±0.15 |
| DEAL\FN | 1.57±0.02 | 2.79±0.02 | 4.03±0.03 |
| EASI | 2.94±0.85 | 4.64±1.50 | 6.03±1.82 |
| BO | 4.67±0.01 | 7.29±0.01 | 10.06±0.01 |

Table 14: Average search error percentage($\times 100\%$).

| | AllegroHand | | |
|---|---|---|---|
| Method | $\xi =10$ | $\xi =15$ | $\xi =20$ |
| DEAL | **0.83±0.02** | **1.69±0.03** | **2.57±0.05** |
| DEAL\PN | 2.60±0.03 | 3.80±0.05 | 4.92±0.05 |
| DEAL\FN | 1.59±0.01 | 2.80±0.17 | 4.06±0.20 |
| EASI | 4.25 ± 0.12 | 6.82±0.20 | 9.32±0.25 |
| BO | 3.75 ±0.01 | 6.32±0.03 | 9.16±0.06 |

