# OpenReview forum: "DEAL: Diffusion Evolution Adversarial Learning for Sim-to-Real Transfer"
_NeurIPS.cc/2025/Conference — NeurIPS 2025 poster_

### Official Review · Reviewer_oQUS · 2025-06-16

**Clarity:** 1
**Significance:** 2
**Originality:** 2
**Rating:** 4
**Confidence:** 3

**Summary:**

This work addresses the problem of system identification for Sim-to-Sim and Sim-to-Real transfer. The authors propose a novel method, DEAL, building upon Diffusion Evolution and Generative Adversarial Networks (GANs). A discriminator is trained to distinguish between two distributions of system parameters: a prior distribution and a target distribution. System identification is then framed as the optimization problem of finding parameter values that fool the trained discriminator. Diffusion Evolution is used to solve this optimization, forming the basis of the DEAL algorithm: a sampling-based method in which the discriminator provides a fitness signal guiding the samples toward high-likelihood regions. An adaptive sampling strategy is introduced, tuning the exploration noise based on fitness. The method is evaluated on several reinforcement learning environments, first through parameter estimation error in Sim-to-Sim setups, then on downstream control tasks in Sim-to-Sim and Sim-to-Real settings. DEAL is compared to several baselines and outperforms them.

**Questions:**

1. In Figure 4, how can DEAL outperform the oracle? Does the oracle use the true parameters with a suboptimal policy?
2. Could the authors elaborate on the benefits of their deep-learning-based approach compared to traditional system identification algorithms?
3. What are the computational and data requirements of DEAL? How do they compare with those of the baselines?
4. What is the architecture of the discriminator network?
5. Did the authors consider using active learning during data collection to improve sample efficiency?

**Ethical Concerns:**

["NO or VERY MINOR ethics concerns only"]

**Final Justification:**

The authors clarified the mathematical formalism in the rebuttal, and addressed my concerns regarding traditional system identification techniques.

I still believe that there is room for improvement in the experimental evaluation.

**Limitations:**

The limitations are discussed in the appendix, but would deserve a dedicated paragraph in the main body. See my suggestions in the 'Strength and Weaknesses' section.

**Quality:**

2

**Strengths And Weaknesses:**

Globally, the idea of the submission is original and relevant. However, the paper suffers from several clarity issues, particularly in the formulation of the problem and the description of the methodology. This makes it difficult to fully assess the contributions and how they improve upon classical baselines.

### Strengths

- Modeling system identification as a generative modeling problem is an interesting and promising direction.
- The method is evaluated across several control environments, with results shown both in terms of parameter estimation and downstream control performance.
- The experimental results are encouraging.

### Weaknesses

**Problem formulation**

The mathematical problem is not properly introduced, and the paper lacks rigor in its formalism.

The paper would benefit from a section that clearly formulates the learning problem, including the objective function the proposed method is designed to optimize, independent of a trained discriminator. Typically, system identification aims to minimize some estimation error (e.g., between true and predicted parameters) or improve control performance. Such a formulation would clarify the purpose of the approach and how it differs from the baselines.

**Method structure**

Section 3 lacks a clear structure and separation between the different phases of the method. Several important points remain ambiguous:

- How is the discriminator trained and update? From (5), Algorithm 1., and the experiments, it seems to be trained without a generator. Is the source distribution sampled by running simulations with parameters drawn uniformly?
- Equation (6) is confusing. Does it define an optimization over fixed discriminator weights, or are these updated jointly with sampling? The similarity to (4) also makes the role of $\theta$ unclear, as it denotes physical parameters while such a formulation is typically used  to denote neural network weights for GANs.
- The statement in lines 46–47 that Diffusion Evolution has a "dual role" is unclear. In what sense does it serve as both a generator and an optimizer?
- The link between discriminator training, the objective in (6), and the final DEAL algorithm (Algorithm 1) is not clearly described.

**Evaluation**

The evaluation process of the proposed algorithm should be stated more thoroughly. It would be clearer if the evaluation metrics were clearly stated in terms of an error function of the true parameters, both for evaluation in parameter space and for evaluation on downstream control tasks.


**Computational cost and data efficiency**

The method’s data and computational efficiency are unclear. In particular:

- DEAL requires training a discriminator on samples from the real-world environment. Training such a network typically requires a substantial amount of data, especially if training happens at each iteration of a search algorithm as in Algorithm 1.
- Section 4.3 is particularly confusing: what does a demonstration of length 1 mean? How is the discriminator trained in such a low-data regime?
- The introduction claims that DEAL "avoids the need for costly real-world data," yet the method depends on real samples to train the discriminator. This contradiction should be addressed.
- Computational cost and sample complexity are key in Sim-to-Real applications. In addition to Table 3, the paper should report training times and simulation budget for DEAL and the baselines, and discuss any trade-offs. The limitations of the proposed approach in direction should be discussed in the main body of the paper.

**Conceptual justification**

The use of diffusion models for system identification is an original idea, but the motivation is weakly supported:

- Why aren’t classical methods such as least squares regression on physical parameters considered? With access to a simulator, least squares or maximum likelihood estimation seems natural.
- Is the method meant to work in a model-free setting? If so, this should be clearly stated and contrasted with model-based approaches.
- More generally, modeling system identification as an adversarial sampling problem needs better theoretical and practical justification. Equation (6) should be compared and contrasted with more standard formulations in the literature.

**Minor clarity issues**

- In line 169, parameters are described as Gaussian-distributed, but iterative sampling algorithms such as diffusion evolution  are usually designed to represent complex, non-Gaussian distributions. This should be clarified.
- Equation (8) is referenced before it is introduced.
- There are several typos throughout the paper that hinder readability.

---

> ### Author Rebuttal · Authors · 2025-07-31
>
> `W1 Problem formulation`
>
> `A1.`We appreciate your feedback and would like to clarify the process of our method for you.
>
> The goal of DEAL is to minimize the discrepancy between state transition distributions from different domains as follows:
> $$
> \mathop{\min}\limits_{\theta\in\mathcal{U}}\Vert\mathcal{P}_t(\theta^{target}),\mathcal{P}_s(\theta)\Vert.
> $$
>
> Our framework is based on GANs, where the generator and discriminator are trained jointly in an adversarial manner. Accordingly, DEAL also aims to minimize the discrepancy through the joint training of generator and  discriminator. Notably, our generator DE does not involve neural networks, it generates and optimizes the physical parameter distribution $\theta$ to participate in adversarial training with the discriminator.
>
> Therefore, **$\theta$ and the discriminator are optimized jointly, rather than training the discriminator first and then optimizing $\theta$.** To improve clarity, we can express the trajectory generated in the simulator as $d^{\mathcal{S}}(\theta,\pi_0)$ and the trajectory pre-collected in the real world as  $d^{\mathcal{T}}(\theta^{target},\pi_0)$, then the objective function Equation (6) can be written as follows:
>
> $$
> \theta^*=\mathop{\arg\min}\limits_{\theta\in\mathcal{U}}\max\limits_{D}\mathbb{E}_ {(\mathbf{s},\mathbf{a},\mathbf{s}')\sim d^{\mathcal{T}}(\theta^{target},\pi_0)}[D(\mathbf{s},\mathbf{a},\mathbf{s}')]-\mathbb{E}_{(\mathbf{s},\mathbf{a},\mathbf{s}')\sim d^{\mathcal{S}}(\theta,\pi_0)}[D(\mathbf{s},\mathbf{a},\mathbf{s}')] .
> $$
>
> This formulation makes it clear that the overall objective in Equation (6) describes the joint optimization of $\theta$ and the discriminator, the trajectories from both domains are jointly sampled and input to the discriminator for scoring.
>
> Since we are optimizing over the physical parameter distribution $\theta$ rather than a generator neural network, this may lead to misunderstanding.
> Thank you for your question and we have refined the problem formulation in the revised version.
>
> `W2 Method structure`
>
> `A2.`
>
> 1. DEAL follows the same paradigm as GANs, where **the discriminator is not pre-trained separately but co-optimized with the generator in an interactive manner.** In Equation (5), the source-domain data $d^{\mathcal{S}}$ depends on physical parameters $\theta$ produced by the generator. The inner loop of the discriminator in Algorithm 1 represents epoch-wise training rather than a full pre-training stage. During this interactive process, the source-domain parameters are freshly generated by DE at each step.
> 2. **Equation (6) formalizes the joint adversarial optimization of the discriminator and parameter distribution given by DE, rather than optimizing over a fixed, pre-trained discriminator.** In contrast to Equation (4), where the generator learns neural network weights, DEAL performs optimization over a distribution of physical parameters that influence the trajectory, since DE does not involve any neural networks internally.
> 3. DE plays a dual role as a diffusion-based generator that produces parameters to configure the simulator while simultaneously optimizing these parameters based on the discriminator's feedback.
> 4. To summarize, DEAL does not include separate discriminator pre-training. Equation (6) defines the overall objective of DEAL. The discriminator's objective is embedded in Equation (6), DE generates and optimizes parameters to adversarially minimize this state-transition distribution gap.
>
> `W3 Evaluation`
>
> `A3.`The evaluation metric in Sections 4.1 and 4.3 is the average percentage error of the searched parameters relative to the true parameters, defined as follows:
> $$
> \epsilon_p = \mathbb{E}\left(\left|\frac{{\theta} - \theta^{target}}{\theta^{target}}\right|\right) \times 100\\%
> $$
> Thank you for your suggestion. We have included clear evaluation formulas in the revised version.
>
> `W4 Computational cost and data efficiency / Q3`
>
> `A4.`
>
> 1. As DEAL does not require a pre-trained discriminator at each iteration. The discriminator and parameter distribution $\theta$ are updated jointly, so while some trajectory data is needed, the amount is not excessive.
> 2. In Section 4.3, the humanoid task performs sim2sim parameter search using trajectories collected from 200 parallel environments, as mentioned in Experiment Details. Therefore, the real trajectory here refers to 200 = (parallel number = 200)$\times$(trajectory length = 1) state transitions in total.
> 3. In line 52, this phase is followed by "to train a noise prediction module", indicating it only refers to not using real samples for training the noise predictor. Since DE simulates a denoising process without involving a diffusion process to train the noise predictor.
> 4. Because DEAL does not separately pretrain the discriminator, the time reported in Table 3 represents the combined training cost of both the parameter distribution and the discriminator.
>
> Additionally, we provide the simulation data budgets for each task, all tasks run 200 parallel environments, but the trajectory lengths differ across tasks. The detailed numbers are shown in the table below.
>
> |Data budget|Ant|CartPole|Humanoid|Go2|AllegroHand|
> |-|-|-|-|-|-|
> |For DEAL and baselines|200*200|200*200|200*200|200*250|200*600|
>
> To compare the data requirements of DEAL and other baselines, we collect between $\eta$=1 and $\eta$=200 trajectories and do parameter search in Humanoid task, then we measure the average search error to evaluate each method's data efficiency.
>
> |Avg. Search Error (%)|$\eta$ = 1|$\eta$ = 5|$\eta$ = 10|$\eta$ = 50|$\eta$ = 200|CPU Memory(G)|GPU Memory(G)|
> |-|-|-|-|-|-|-|-|
> |DEAL|12.5±4.4|13.8±5.0|13.6±5.2|13.9±5.1|12.9±4.7|2.3|0.7|
> |EASI|66.0±20.2|54.1±16.4|57.8±20.2|54.2±19.0|58.2±19.4|2.4|0.9|
> |BO|108.5±36.9|107.8±36.7|107.6±37.0|102.5±36.6|97.7±37.1|2.4|0.9|
>
> The table above shows that **DEAL achieves the best optimization results using the same amount of trajectory, confirming that DEAL requires less data than other baselines.**
>
> `W5 Conceptual justification`
>
> `A5.`
> 1. Due to the non-differentiability of simulators, using least squares or maximum likelihood to compute trajectory gradients doesn't allow backpropagation to physical parameters. Moreover, as the parameter space grows exponentially, gradient-based optimization becomes increasingly intractable. In contrast, DEAL leverages the high-dimensional adaptability of diffusion models to scale more efficiently.
> 2. We compare DEAL with Model-based method Extended Kalman Filter(EKF) on the CartPole task using the same trajectory data. As shown in the table, EKF suffers from significant bias due to linearization errors when estimating nonlinear parameters.
> |Avg. Search Error (%)|CartPole|
> |-|-|
> |DEAL|6.9±1.1|
> |EKF|37.4±8.8|
> 3. Thank you for your suggestion. We have refined and clarified Equation (6).
>
> `W6 Minor clarity issues`
>
> `A6.`In DEAL, population individuals iteratively evolve from initial noise, forming a complex distribution that ultimately converges near target parameters. For improved sampling, we fit a Gaussian to the final population's mean and variance, using it as the retraining distribution.
>
> We appreciate your feedback and have addressed other clarity issues in the revised version.
>
> `Q1. In Figure 4, how can DEAL outperform the oracle?`
>
> `A7.`When the target parameters are not sufficiently well set, training in the target domain can become relatively more difficult. For example, in tasks like Ant or AllegroHand, excessive mass or weak actuators may slow down oracle training. The searched parameter distribution, due to its bias, may sample easier parameters during training, which can accelerate learning and temporarily result in higher rewards than the oracle. Such fluctuations tend to diminish as training converges, without affecting the oracle’s theoretical optimality.
>
> `Q2. Could the authors elaborate on the benefits of their deep-learning-based approach compared to traditional system identification algorithms?`
>
> `A8.`DEAL does not rely on linear assumptions or explicit gradient access often required by traditional methods. Instead, it performs sample-based blackbox optimization, making it broadly applicable to nonlinear systems. Unlike traditional methods that require covariance estimation in high-dimensional spaces, DEAL avoids numerical instability by directly executing denoising updates via Eq. (9). Through discriminator-guided reweighting in Eq. (1), it efficiently identifies high-fitness regions, effectively leading to more targeted and stable updates. The diffusion process further enables adaptive step size control, allowing DEAL to stably and rapidly converge toward the optimal solution.
>
> `Q4. What is the architecture of the discriminator network?`
>
> `A9.`In this work, the discriminator is implemented as a fully connected MLP with input $(\mathbf{s},\mathbf{a},\mathbf{s}')$, two hidden layers of 256 units with ReLU activation, and a scalar output. Furthermore, We adopt the WGAN framework to address instability during discriminator training.
>
> `Q5. Did the authors consider using active learning during data collection to improve sample efficiency?`
>
> `A10.`We incorporate active learning (AL) for data collection in both Ant and CartPole tasks, one UDR policy is trained with AL by jointly learning an uncertainty predictor to guide exploration toward more informative trajectories, while the other is trained without AL. Each policy is then used to collect a single trajectory of length 200 for parameter search. After retraining new policies with the respective optimized parameters and deploying them in the target domain, we compare their transfer return. As shown in the table below, the results indicate that active learning improves sample efficiency by embedding more target-domain information into trajectories of the same length, thereby enhancing policy adaptability.
> |Transfer return|Ant|CartPole|
> |-|-|-|
> |DEAL with AL|700.5±29.4|183.8±4.1|
> |DEAL|637.3±37.7|176.6±4.0|

---

> > ### Comment · Reviewer_oQUS · 2025-08-04
> >
> > Thank you for your detailed response.
> >
> > These clarifications, particularly those concerning the problem setup, improve understanding the relervance of the method. I strongly encourage the authors to write a clear mathematical formulation of the problem under study in a revised version.
> >
> > Regarding the conceptual justification of DEAL, the authors' response clarified the relevance of a black-box method over traditional gradient-based system identification techniques. However, I still believe that it would be insightful to compare DEAL to a least-squares estimation baseline in the case of a simple robotic system with non-linear dynamics, for which the simulator can be differentiated through. This could help assess the improvement of diffusion evolution in finding the optimal parameters in comparison with a non-convex optimization problem formulation, in a controlled scenario.
> >
> > As mentioned by reviewer wvcr, the results highly depend on the quality of the real-world samples used to train the discriminator. I also agree with their point regarding the method novelty.
> >
> > I increase my score to 4.

---

> > > ### Author Response · Authors · 2025-08-05
> > >
> > > Dear Reviewer oQUS:
> > >
> > > We greatly appreciate your suggestions and will incorporate them into the next revision of the paper. In addition, we have conducted several experiments to further address your concerns.
> > >
> > >
> > > `1.` We include an experiment on the CartPole system using least-squares estimation. Although this system is relatively simple in our paper, it constitutes a nonlinear system governed by 11 physical parameters. The comparison results are shown in the table below:
> > >
> > > |Avg. Search Error (%)|CartPole|
> > > |-|-|
> > > |DEAL|6.9±1.1|
> > > |EKF|37.4±8.8|
> > > |Least-squares|53.0±19.6|
> > >
> > > The least-squares estimation exhibits relatively large search errors in this nonlinear system, and the results are highly sensitive to random initialization. In contrast, DEAL demonstrates stable and accurate search performance in such nonlinear systems.
> > >
> > > `2.` To address the concern regarding DEAL's reliance on trajectory quality, We conduct experiments in the Humanoid and CartPole environments using 200-steps trajectories collected from policy checkpoints at different stages of training. These trajectories are then used for parameter search, allowing us to evaluate how the quality of the trajectory affects DEAL's performance. The results are presented in the table below:
> > >
> > > | Iterations（CartPole） | Avg. Search Error (%) |
> > > | ---------------------- | --------------------- |
> > > | 50                     | 8.84 ± 2.96          |
> > > | 100                    | 7.87 ± 2.35          |
> > > | 250                    | 7.19 ± 2.96          |
> > > | 1000                   | 7.34 ± 2.88          |
> > > | 2500                   | 5.10 ± 2.02          |
> > >
> > > | Iterations（Humanoid） | Avg. Search Error (%) |
> > > | ---------------------- | --------------------- |
> > > | 5e3                    | 13.59 ± 5.37         |
> > > | 1e4                    | 14.70 ± 4.80         |
> > > | 2e4                    | 13.08 ± 5.00         |
> > > | 2.5e4                  | 10.24 ± 4.13         |
> > > | 3e4                    | 10.78 ± 3.40         |
> > >
> > > **Contrary to concerns about its reliance on trajectory quality, DEAL demonstrates strong robustness and can still accurately search parameters even when using low-quality trajectories collected from early-stage policies.** As training progresses and trajectory quality improves, DEAL's search performance improves accordingly.
> > >
> > > Once again, we sincerely thank you for your time and insightful feedback.

---

> > > > ### Comment · Reviewer_oQUS · 2025-08-05
> > > >
> > > > Thank you for conducting and sharing these additional experiments. The new results strengthen the paper’s empirical contributions and  improve the overall impact of the work.
> > > >
> > > > That said, I still have reservations regarding the overall clarity of the paper, and the level of conceptual novelty For this reason, I maintain my updated score of 4 from the previous round.

---

> > > > > ### Author Response · Authors · 2025-08-05
> > > > >
> > > > > Dear Reviewer oQUS:
> > > > >
> > > > > Thank you once again for your insightful comments and helpful suggestions, they have been very enlightening for us. If you have any further questions, please contact us.
> > > > >
> > > > > Sincerely, thank you for your time.

---

### Official Review · Reviewer_wvcr · 2025-06-26

**Clarity:** 3
**Significance:** 2
**Originality:** 2
**Rating:** 4
**Confidence:** 3

**Summary:**

This paper introduces DEAL (Diffusion Evolution Adversarial Learning), a novel system identification (Sys-Id) framework designed to close the sim-to-real gap for reinforcement learning policies. The core idea is to align the physical dynamics of a simulator with a target real-world environment by accurately identifying the simulator's physical parameters. DEAL employs a dual mechanism: diffusion evolution and adversarial learning. The discriminator's output serves as a fitness score to guide the DE process, which iteratively refines a population of parameter sets to find an optimal configuration that minimizes the discrepancy between simulation and reality. The authors validate DEAL on a range of sim-to-sim and sim-to-real robotics tasks, demonstrating its superior performance in high-dimensional parameter identification and sim-to-real transfer with limited real-world data compared to several baselines.

**Questions:**

1. How does the framework handle situations where the real-world dynamics are non-stationary? Could DEAL be adapted for online or continuous system identification?
2. Did you encounter common stability issues in Eq. (6) with UDR?

**Ethical Concerns:**

["NO or VERY MINOR ethics concerns only"]

**Final Justification:**

After carefully reading the rebuttal, I have decided to maintain my original score.

**Limitations:**

yes.

**Paper Formatting Concerns:**

no concern.

**Quality:**

3

**Strengths And Weaknesses:**

## Strengths

* The primary strength of this work is its demonstrated ability to successfully perform system identification in high-dimensional parameter spaces.
* The method is explicitly designed to work with only a small amount of few-shot real-world data. The experiments confirm that DEAL can achieve high identification accuracy with sparse trajectories.
* The combination of Diffusion Evolution and adversarial learning appears to be an appealing approach to the parameter searching for reinforcement learning controllers.

## Weaknesses
* **Novelty.** While the combination of the components is novel, the underlying techniques are not. The paper leverages existing concepts, and the core contribution is the integration of these parts into a new framework for a specific problem, rather than a fundamental algorithmic breakthrough.
* **Computational Complexity.** The overall framework still requires iteratively running a large number of parallel simulations (pp. 7), training a discriminator network, and executing the evolutionary search. Although the authors state the search itself is fast, the total computational cost of the pipeline (including initial policy training, data collection, and final policy retraining) could be substantial, potentially limiting its accessibility.
* The method's success hinges on the quality of the initial real-world trajectories used to train the discriminator. These trajectories are collected using a policy trained with uniform domain randomization. If this policy performs extremely poorly and fails to exhibit informative behaviors in the real world, it may not provide a strong enough signal for the discriminator, potentially limiting the effectiveness

---

> ### Author Rebuttal · Authors · 2025-07-31
>
> `Q1. Computational Complexity.`
>
> `A1.` The training cost of the policy itself is not included in our method. Instead, by searching for parameters using DEAL, **the need for manual tuning and policy training time is significantly reduced compared to UDR,** because after parameter search, the domain knowledge that the policy needs to acquire is of high precision and quality. Moreover, if the new parameters remain within the original bounds, we only need to fine-tune the policy.
>
> **Data‑collection cost is also minimal in DEAL.** As described in Section 4.4's Go2 Sim2Real experiment, only a single real robot was run for 20 seconds to collect a trajectory containing 1,000 state transitions for parameter search. Furthermore, if DEAL is only used to search for static parameters, such as mass or motor properties, which are intrinsic physical parameters of the robot, the identified parameters can be reused across different controllers and environments, leading to substantial savings in both computational and time costs.
>
> `Q2. The method's success hinges on the quality of the initial real-world trajectories used to train the discriminator. `
>
> `A2.`  Current learning-based system identification methods do require the collection of trajectories with a certain quality in the real world. However, we believe that even trajectories from very poor policies still encode physical dynamics information. When using the same policy to search parameters, the failed behavior should be reproduced by adjusting those parameters.
>
> To verify this, we train the  UDR policies for the CartPole and Humanoid in simulation, and take the policies at various training checkpoints to collect a 200-step trajectory in the target environment. We then search for parameters within $[\tfrac{1}{3}\times\theta^{target},3\times\theta^{target}]$ to reflect the impact of the policies performance on the search results. In the tables, 50 denotes the policy after 50 training iterations, and the policy only starts to converge in the last row.
>
> | Iterations（CartPole） | Avg. Search Error (%) |
> | ---------------------- | --------------------- |
> | 50                     | 8.84 ± 2.96          |
> | 100                    | 7.87 ± 2.35          |
> | 250                    | 7.19 ± 2.96          |
> | 1000                   | 7.34 ± 2.88          |
> | 2500                   | 5.10 ± 2.02          |
>
> | Iterations（Humanoid） | Avg. Search Error (%) |
> | ---------------------- | --------------------- |
> | 5e3                    | 13.59 ± 5.37         |
> | 1e4                    | 14.70 ± 4.80         |
> | 2e4                    | 13.08 ± 5.00         |
> | 2.5e4                  | 10.24 ± 4.13         |
> | 3e4                    | 10.78 ± 3.40         |
>
> **The tables demonstrate that DEAL is capable of identifying reasonably accurate parameters, even when starting from a poorly performing policy.**  Naturally, as the controller's performance improves, the quality of the generated trajectories increases significantly, leading to search results that more closely match the ground truth.
>
> `Q3. How does the framework handle situations where the real-world dynamics are non-stationary? Could DEAL be adapted for online or continuous system identification?`
>
> `A3.`
>
> 1. In non-stationary environments, physical parameters may continuously change, and the collected samples will be multimodal. Since the discriminator can only perform binary classification, it cannot generate effective signals for DE in such cases. However, we believe that future work will develop more general discriminators to match our generator, enabling the accurate identification of non-stationary environment.
> 2. In online system identification, parameters found by DEAL can directly boost a model-based controller. But for model-free methods like reinforcement learning, fine-tuning in the updated simulator is needed, which isn't real-time. Therefore, all comparisons in this paper are based on offline parameter search. However, for underactuated mobile robots, we usually focus on identifying key internal physical parameters (e.g., joint motor coefficients and mass) that rarely change significantly and are crucial for sim2real transfer. To better adapt to changes in the external environment, we use the RMA controller with an adaptation module to handle environmental variations. This approach complements DEAL's parameter search and significantly improves transfer performance.
>
> `Q4. Did you encounter common stability issues in Eq. (6) with UDR?`
>
> `A4.` When training vanilla GANs, we encountered stability issues. WGAN[1] has proved that vanilla GANs minimize the JS divergence between distributions, which becomes constant when distributions have no overlap, causing gradient vanishing and unstable training. To address this, we switched to WGAN, which uses the Wasserstein distance to measure distribution gaps. Unlike JS divergence, Wasserstein distance remains effective even when distributions don't overlap. We also applied gradient penalty or weight clipping to enforce Lipschitz continuity and ensure valid Wasserstein distance computation.
>
> [1] Arjovsky, Martin, Soumith Chintala, and Léon Bottou. "Wasserstein generative adversarial networks." International conference on machine learning. PMLR, 2017.

---

### Official Review · Reviewer_pNN9 · 2025-07-05

**Clarity:** 2
**Significance:** 2
**Originality:** 2
**Rating:** 4
**Confidence:** 3

**Summary:**

RL has demonstrated significant success when trained in simulation, providing cost-efficient and safety advantages. However, when these policies learned in simulation are deployed in the real world, they often experience a reality gap. This paper presents a solution to this problem by utilizing concepts from GANs and diffusion models, introducing a method called Diffusion Evolution with Adversarial Learning (DEAL). Specifically, the paper proposes a new GAN-style framework for system identification, aiming to enhance the simulator and narrow the gap between real-world deployment and simulation. The paper conducted experiments across various tasks, ranging from CartPole (a classic control task) to Humanoid, and showed that the policy learned with the improved simulator dynamics is capable of matching the performance of the policy trained with oracle dynamics.

**Questions:**

- What is the formula for simulator function that theta is optimizng or apart of? For exmaple, in [1], theta is an extra parameter of the learning dynamics.
- In the GAN formula in equation 6, what variables in the equation depend on theta?
- If you are strictly matching the the data distrubiton (s,a,s') from d^t and d^s; how you you decouple the policies being different from the dynamics being different?
- In DEAL, are you jointly learning a policy, simulator parameters, and a discriminator, and then using the learned dynamics to train a new policy?
- What are the parameters in the simulators you're trying to learn? How do you decide on these parameters to learn?



[1] SimGAN: Hybrid Simulator Identification for Domain Adaptation via
Adversarial Reinforcement Learning

**Ethical Concerns:**

["NO or VERY MINOR ethics concerns only"]

**Final Justification:**

The author's rebuttal addressed my concerns about confusion regarding certain concepts in the paper, and as a result, I raised my score.

**Limitations:**

Yes

**Quality:**

2

**Strengths And Weaknesses:**

Strengths:

Reducing the reality gap when training a policy in simulation and deploying it in the real world is very important. The paper's perspective of framing this problem as a GAN and employing modern models to assist in learning the simulator's parameters makes sense.

Weaknesses:

The paper's writing makes it somewhat challenging to follow. It never clearly defines the simulator parameter theta, leaving it unclear whether this parameter is part of the transition dynamics or something else. The overall concept of the GAN formulation is understandable, but it is not clear how the other parameters relate to theta. While it is obvious how you are optimizing theta using diffusion, it is not clear what exactly is being optimized.

---

> ### Author Rebuttal · Authors · 2025-07-31
>
> `Q1. The clear definition of the simulator parameter theta`
>
> `A1.` In this paper, **the specifics of $\theta$ are detailed in  Section 4.1 and Appendix A.5. It refers to the physical dynamic parameters of the robot or external environment**, such as mass, length and motor coefficients, which directly affect the state transition distribution:  $s_{t+1}\sim p(s_{t+1}|s_{t},a_{t};\theta)$. We optimize $\theta$ in the parameter space by using DE to make the corresponding simulated state transition more similar to the real world state transition, and the discriminator is updated jointly in this process, the objective function Equation (6) can be rewritten as follows:
>
> $$
> \theta^*=\mathop{\arg\min}\limits_{\theta\in\mathcal{U}}\max\limits_{D}\mathbb{E}_ {(\mathbf{s},\mathbf{a},\mathbf{s}')\sim d^{\mathcal{T}}(\theta^{target},\pi_0)}[D(\mathbf{s},\mathbf{a},\mathbf{s}')]-\mathbb{E}_{(\mathbf{s},\mathbf{a},\mathbf{s}')\sim d^{\mathcal{S}}(\theta,\pi_0)}[D(\mathbf{s},\mathbf{a},\mathbf{s}')],
> $$
>
> where $d^{\mathcal{S}}(\theta,\pi_0)$ represents the trajectory collected by $ \pi_0$ in the simulator parameterized by $\theta$. In this adversarial process, DE acts as the generator to generate and optimize the $\theta$, making the simulation trajectory increasingly similar to the real world trajectory  $d^{\mathcal{T}}(\theta^{target},\pi_0)$ from the discriminator's perspective,  while the discriminator aims to distinguish between them as much as possible.
>
> Thank you for raising this issue. We have added a clear definition of $\theta$ in Section 3 of the revised version to facilitate your understanding.
>
> `Q2. What is the formula for simulator function that theta is optimizng or apart of?`
>
> `A2.` $\theta$ refers to the physical parameters of the robot or external environment. It is not the weight of neural networks or a parameter of function. **The formula for simulator optimization related to $\theta$ is Equation (6) in answer A1.** In DEAL, we use DE to denoise and optimize $\theta$ to minimize the discrepancy between the real and simulated state transition distributions as perceived by the discriminator.
>
> `Q3. In the GAN formula in equation 6, what variables in the equation depend on theta?`
>
> `A3.` In Equation (6) from answer A1, **the embedded GAN objective function includes simulated trajectories $d^{\mathcal{S}}(\theta,\pi_0)$, which depend on $\theta$.** We optimize $\theta$ using the generator DE and thereby parameterize the simulator, which in turn affects the trajectories collected from the simulator. These trajectories are then used in adversarial training against the discriminator, since the discriminator is always trying to distinguish between simulated and real-world trajectories.
>
> `Q4. How do you decouple the policies being different from the dynamics being different?`
>
> `A4.` **We use the same policy when collecting real-world trajectories and when searching for parameters or collecting simulated trajectories.** This ensures that the parameters we search for are solely aimed at matching the dynamics of the real-world environment.
>
> Specifically, we first train an initial sub-optimal policy $\pi_0$ in simulation based on UDR, and then deploy it in the real environment to collect offline trajectories $d^{\mathcal{T}}(\theta^{target},\pi_0)$. During the subsequent parameter search, we continue to use $\pi_0$ to collect trajectories from the simulator parameterized by $\theta$, denoted as $d^{\mathcal{S}}(\theta,\pi_0)$. The discriminator then samples from both datasets and assigns scores to distinguish them, updating its network accordingly, while DE updates the distribution of $\theta$ based on the discriminator's evaluation.
>
> `Q5. In DEAL, are you jointly learning a policy, simulator parameters, and a discriminator?`
>
> `A5.` No, these three components are not trained jointly. The sub-optimal policy $\pi_0$ is pre-trained via UDR, and its weights are frozen during the parameter search as shown in Figure. 1. Therefore, **only the simulator parameters $\theta$ and the discriminator are updated during the search process.** This procedure eventually converges to a set of physical parameters that approximate the real-world dynamics, which are then used to reconfigure the simulator and retrain a new policy which better adapted to the real world.
>
> `Q6. What are the parameters in the simulators you're trying to learn? How do you decide on these parameters to learn?`
>
> `A6.` **The parameters contained in $ \theta$ are listed in Appendix A.5.** These mainly include the parameters that were randomized during the UDR training process, such as the joint PD gains for the Go2 robot or the pole length and mass in the cartpole task. These are key factors affecting sim-to-real transfer, and our goal is to identify more accurate distributions for them, in order to reduce the computational cost and performance degradation introduced by UDR.

---

> > ### Author Response · Authors · 2025-08-05
> >
> > Dear Reviewer pNN9:
> >
> > Thank you once again for your insightful comments and helpful suggestions. As the deadline for author-reviewer discussions is approaching, we would greatly appreciate it if you could let us know whether our rebuttal has resolved your concerns. Your insights are invaluable to us, and we're eager to address any remaining issues to improve our work.
> >
> > Thank you very much for your time.

---

> > > ### Comment · Reviewer_pNN9 · 2025-08-08
> > >
> > > Thank you for your response to my questions, and I will increase my score.

---

> > > > ### Author Response · Authors · 2025-08-08
> > > >
> > > > Dear Reviewer pNN9:
> > > >
> > > > Thank you for your detailed questions and suggestions，they have been very enlightening for us. If you have any further questions, please contact us.
> > > >
> > > > Sincerely, thank you for your time.

---

### Official Review · Reviewer_rbd9 · 2025-07-09

**Clarity:** 4
**Significance:** 2
**Originality:** 2
**Rating:** 4
**Confidence:** 3

**Summary:**

This paper addresses the problem of  System Identification by introducing a novel method that combines Diffusion Evolution (DE) with Adversarial Learning (AL). The objective is to close the gap between simulator (source) and real-world (target) distributions. The proposed approach follows an iterative process: DE is used as a generator, then a policy trained via UDR is executed in both source and target environments. The discriminator then evaluates the similarity between the two domains using those rollouts. A noise scheduling strategy is introduced to balance exploration and exploitation. The method is evaluated on a wide range of environments, covering both sim-to-sim and sim-to-real tasks, and compared against several baseline approaches.

**Questions:**

- The paper presents tasks where the DEAL framework outperforms SOTA. But are there cases or scenarios where it underperforms ?
- Running rollouts on target environment at each iteration may be costly. How would modify your approach to reduce reliance on real-world rollouts ? What happens if only a fixed set of real-world data is available ? Which is often the case in practice

As I found the main conceptual contribution a bit light, I am assigning a score of 3. But if authors can convincingly address my questions, I would be more than happy to raise my score.

**Ethical Concerns:**

["NO or VERY MINOR ethics concerns only"]

**Final Justification:**

Authors have addressed most of my concerns, which makes me happy to raise my score to 4

**Limitations:**

yes

**Quality:**

3

**Strengths And Weaknesses:**

Strengths

- The paper is clearly written and easy to follow
- Experiments cover both Sim-to-Sim and Sim-to-Real tasks.
- DEAL appears to outperform SOTA methods in these tasks

Weaknesses

- The idea of computing a similarity measure based on collected rollouts on source and domain to optimize a GAN-style process is not new (eg SimGAN).
- The main conceptual novelty of this paper is to use a Diffusion Evolution process as a generator, which may be a relatively modest contribution. Beyond empirical performance, is there any theoretical justification for using DE in terms of convergence or sample efficiency ?
- I did not find detailed description of the discriminator in the paper.  What architecture is used ? How is it regularized to avoid overfitting ?

---

> ### Author Rebuttal · Authors · 2025-07-31
>
> `Q1.The idea of computing a similarity measure based on collected rollouts on source and domain to optimize a GAN-style process is not new (eg SimGAN).`
>
> `A1.` We appreciate the inspiration provided by SimGAN. DEAL introduces several key modifications to address known issues in GAN-style system identification.
>
> 1. Prior approaches focused solely on comparing the similarity of simulator rollouts to real-world demonstrations, with the single goal of making simulated trajectories mirror real-world demonstrations as closely as possible. In contrast, our approach shifts the focus: we employ a discriminator to assess the similarity of physical state transitions, aiming to fundamentally bridge the gap between simulation and reality. This not only enables more efficient use of real-world data but also leads to improved performance.
>
> 2. Previous methods (such as SimGAN) adopt the traditional GAN architecture, using a neural network generator to produce simulator parameters for optimization. However, GAN training is highly unstable, and training the neural generator is time-consuming, requiring significant effort in tuning and debugging. In our work, we introduce Diffusion Evolution (DE) as an alternative for parameter search, which significantly accelerates simulator parameter optimization. Moreover, the combination of DE and the discriminator offers high training stability, avoiding the instability issues commonly encountered in traditional GAN-based methods.
>
> `Q2. Is there any theoretical justification for using DE in terms of convergence or sample efficiency ?`
>
> `A2.` GAN-style parameter search can be viewed as a black-box optimization problem. In DEAL, we choose DE as the generator mainly due to its convergence stability and efficiency, making it more suitable for scaling to complex environments.
>
> Previous black-box optimization methods such as evolution strategies (ES) typically model the parameter distribution as a Gaussian, updating its mean and covariance in each sample iteration. However, the new samples are randomly drawn from the updated distribution and lack explicit control over the convergence trajectory, often resulting in slow convergence and low final accuracy. **Instead of blindly perturbing parameters in each iteration, DE performs a re-weighted posterior estimation of the optimum based on observed samples.** This allows the search process to focus more quickly on regions close to real data distribution.
>
> Specifically, DEAL updates the parameter distribution using the following formulation Eq. (9):
>
> $$
> \theta_{t-1} \gets \sqrt{\alpha_{t-1}}  \hat \theta_0 + \sqrt{1-\alpha_{t-1}-\hat\sigma_t^2}  \hat\epsilon +\hat\sigma_{t}  \omega ,
> $$
>
> where the estimated optimal parameter $\hat\theta_0$ is given by Eq. (1):
>
> $$
> \hat \theta_0(\theta_t,\alpha,t)=\frac{1}{Z}\sum_{\theta \in \Theta_t} \underbrace{g[f(\theta)]} _ {p(\theta_0 = \theta)} \underbrace{\mathcal N(\theta_t; \sqrt{\alpha_t}\theta, 1-\alpha_t)} _ {p(\theta_t | \theta_0 = \theta)}\theta ,  \quad g(x_i) = \frac{e^{x_i}}{\sum_{j} e^{x_j}}.
> $$
>
> In Eq. (1),  $p(\theta_t \mid \theta_0 = \theta)$ is defined by the forward diffusion process: $\theta_t = \sqrt{\alpha_t}\theta_0 + \sqrt{1-\alpha_t}\epsilon$, it acts as a neighborhood weight around parameter $\theta_t$. $p(\theta_0 = \theta)$ represents a probability derived by applying a softmax function $g(\cdot)$ to the discriminator's output, reflecting the likelihood that a state transition under parameter $\theta$ resembles one from the real environment. This formulation effectively uses a softmax-style weighting to emphasize high-fitness regions, guiding $\hat\theta_0$ rapidly toward the true optimal parameters. As a result, the estimate $\hat\theta_0$ serves as a proxy for the gradient direction of the WGAN's objective function $\mathcal{L}$ in Eq. (5) as follows, then the population is updated directionally via Eq. (9) to move toward this optimal estimate.
>
> $$
> \underbrace{\mathcal{L}}_{\searrow} = \mathbb{E} _ {(\mathbf{s},\mathbf{a},\mathbf{s}')\sim d^{\mathcal{T}}(\theta^{target},\pi_0)}[D(\mathbf{s},\mathbf{a},\mathbf{s}')]-\underbrace{\mathbb{E} _ {(\mathbf{s},\mathbf{a},\mathbf{s}')\sim d^{\mathcal{S}}(\theta,\pi_0)}[D(\mathbf{s},\mathbf{a},\mathbf{s}')]} _{\nearrow}.
> $$
>
> **The diffusion noise schedule also helps to stablize the convergence by determining the update step at each iteration.** As $t \to 0$, $\alpha_t \to 1$ and $\sigma_t \to 0$, reducing the variance of $\mathcal{N}(\theta_t; \sqrt{\alpha_t} \theta, 1 - \alpha_t)$ and encouraging fine-grained, local updates in the later stages. The update $\theta_{t-1} \to \hat\theta_0$ serves as a denoising step, similar to the reverse process in diffusion models. With a well-trained discriminator, this process enables stable convergence toward the true optimum.
>
> Additionally, unlike traditional ES that rely on elitism or truncation, which can lose global information and lead to premature convergence to local optima, **DEAL utilizes the entire population in Eq. (1).** It assigns soft weights based on discriminator-derived fitness to estimate the gradient direction of the WGAN's objective function. This leads to more targeted and stable updates, offering improved convergence stability, efficiency and accuracy over previous methods.
>
> We appreciate your insightful question regarding the theoretical motivation behind DE. We have incorporated this detailed analysis into the revision to strengthen the conceptual grounding of our method.
>
> `Q3. The detailed description of the discriminator? How is it regularized to avoid overfitting ?`
>
> `A3.` The discriminator used in DEAL adopts **a fully connected MLP with input $(\mathbf{s},\mathbf{a},\mathbf{s}')$, two hidden layers of 256 units with ReLU activation, and a scalar output**. In DEAL, we employs WGAN [1] , which utilizes Wasserstein distance to measure the distribution discrepancy. When two distributions do not overlap, the Wasserstein distance can still accurately reflect their proximity. Furthermore, we apply gradient penalty or weight clipping to regularize the discriminator, preventing overfitting or model collapse.
>
> Thank you for pointing out this issue. We have included the detailed description of the discriminator in the revised version.
>
> [1] Arjovsky, Martin, Soumith Chintala, and Léon Bottou. "Wasserstein generative adversarial networks." International conference on machine learning. PMLR, 2017.
>
> `Q4. Are there cases or scenarios where DEAL underperforms ?`
>
> `A4.` In current tasks ranging from low-dimensional to high-dimensional scenarios, DEAL significantly outperforms other baselines. However, DEAL also shares some common issues faced by GAN-style methods.
>
> When the target distribution is highly multimodal, existing GAN-style methods may struggle to discriminate among multiple source and target distributions at once, as noted in Appendix A.4 (Limitations). Although diffusion models can capture multimodality,  the discriminator still averages over diverse state transitions rather than converging to distinct solutions. In this case, DEAL tends to homogenize the influence of such multi-source state transitions rather than converge to multiple exact solutions. However, we believe that future work will develop more general and powerful discriminators to match our generator, enabling the simultaneous identification of multiple environment parameter sets.
>
> `Q5. How would modify your approach to reduce reliance on real-world rollouts ? What happens if only a fixed set of real-world data is available ? Which is often the case in practice.`
>
> `A5.` It should be clarified that DEAL uses only offline trajectories. As shown in Alg. 1 and Figure. 1, the policy is not run simultaneously in the simulator and in the real environment during parameter search. In other words, **DEAL itself uses a fixed set of real-world data.** Specifically, we first collect fixed length offline trajectories in the real world using the UDR policy. During parameter search, we only collect trajectories in simulation and sample from the stored real-world trajectories, using the discriminator to distinguish them until their distributions align.
>
> To further reduce the reliance on real data, we introduce an active learning (AL) module. During UDR policy training, we jointly train an uncertainty predictor, which biases policy actions toward more informative trajectories. In the Ant and CartPole task, we collect one real‑world trajectory of length 200 with each of the standard and active learning models. After conducting parameter search for each, we retrain the new policies and evaluate their returns in the target domain. The table shows the result, which demonstrates that active learning indeed improves sample efficiency by embedding more target‑domain information into trajectories of the same length, thereby improving transfer performance.
>
> | Transfer return | Ant              | CartPole       |
> | --------------- | ---------------- | -------------- |
> | DEAL with AL    | **700.5±29.4** | **183.8±4.1** |
> | DEAL            | 637.3±37.7     | 176.6±4.0     |

---

### Note · Authors · 2025-08-13

Dear Area Chair and reviewers:

We thank you for the thorough feedback on our paper. We have carefully addressed all concerns through new experiments, clearer problem statements and contribution analysis.

**Conceptual Justification**
In the rebuttal, we presented a theoretical analysis of employing Diffusion Evolution (DE) as the generator in DEAL, along with a comparative study highlighting DEAL's advantages over traditional black-box optimization methods. These analyses fully articulate the theoretical motivation for adopting DEAL in this work.

**Clarity**
Since DEAL directly optimizes the distribution of physical parameters rather than training a generator network, this may lead to several misunderstandings among reviewers. In the rebuttal, we have addressed each clarity issue and cleared up these concerns.

**Trajectory Quality & Data Budget**
In experiments concerning trajectory quality, we verified that DEAL remains effective even when the policy is in the early stages of training and the trajectory quality is very poor. In addition, under the same data budget, our results show that DEAL achieves the best data efficiency.

**Novelty**
In our discussion, we clarified that DEAL is the first framework to introduce a diffusion-model-based optimizer into high-dimensional system identification, achieving high-precision, high-dimensional, and stable physical parameter identification in nonlinear systems. Moreover, unlike previous GAN-style methods that primarily focus on trajectory approximation, the discriminator in DEAL innovatively evaluates state transitions, thereby focusing more on capturing the system's underlying physical information.

In summary, the rebuttal provided thorough explanations and supplementary experiments addressing the concerns above. The reviewers expressed agreement with our responses and offered positive feedback leaning toward acceptance.

Sincerely, thank you very much for your time.

Best regards,

The authors of Paper 21398

---

### Decision · Program_Chairs · 2025-09-17

**Decision:**

Accept (poster)

**Comment:**

This paper proposes DEAL, a system identification method that combines diffusion evolution with adversarial learning for better sim2real transfer in reinforcement learning. By iteratively refining physical parameters using a discriminator for transition similarity and diffusion evolution for parameter optimization, DEAL achieves higher stability and accuracy in high-dimensional settings. Experiments show it outperforms baselines in sim2real transfer while requiring minimal real-world data. Overall, the results are interesting and have sufficient data to back them up. Please consider some of the presentation details raised by the reviewers (adding more details about the discriminator and training of parameters, policies) in a revision of your draft.